# Structure of lymphostatin, a large multi-functional virulence factor of pathogenic *Escherichia coli*

Matthias Griessmann [1], Tim Rasmussen [1], Vanessa J. Flegler[1], Christian Kraft[1], Ronja Schneider [1], Max Hateley[2], Lukas Spantzel [3], Mark P. Stevens[2], Michael Börsch [3] & Bettina Böttcher [1] ✉

Lymphostatin is a key virulence factor of enteropathogenic and enterohaemorrhagic *Escherichia coli*, playing roles in bacterial colonisation of the gut and in the inhibition of lymphocyte proliferation and proinflammatory responses. The protein's glycosyltransferase and cysteine protease motifs are required for activity against lymphocytes, but high-resolution structural information has proven elusive. Here, we describe the structure of lymphostatin from enteropathogenic *E. coli O127:H6*, determined by electron cryomicroscopy at different pH values. We observe three conformations of a highly complex molecule with two glycosyltransferase domains, one PaToxP-like protease domain, an ADP-ribosyltransferase domain, a vertex domain and a delivery domain. Long linkers hold these domains together and occlude the catalytic sites of the N-terminal glycosyltransferase and protease domains. Lymphostatin binds to bovine T-lymphocytes and HEK-293T cells, forming clusters at the plasma membrane that are internalized. With six distinct domains, lymphostatin can be regarded as a multitool of pathogenic *Escherichia coli*, enabling complex interactions with host cells.

Enteropathogenic and enterohemorrhagic *Escherichia coli* (EPEC and EHEC, respectively) are key causes of paediatric diarrhoea and infections can be life-threatening[1]. They belong to a family of attaching and effacing (A/E) bacteria, which adhere intimately to the apical surface of enterocytes and destroy brush border microvilli. EPEC, non-O157 strains of EHEC, and the murine A/E pathogen *Citrobacter rodentium* almost invariably possess lymphostatin (LifA or Efa1 in EHEC), a 366 kDa protein that plays a pivotal role in intestinal colonisation and has been reported to both inhibit lymphocyte function and mediate bacterial adherence[2–8] without having directly cytotoxic effects. In affinity-purified form, recombinant LifA is active in the femtomolar range against lymphocytes[8,9] and evidence exists that it blocks their proliferation in a manner associated with cell cycle arrest and not with apoptosis or necrosis[10]. Lymphostatin homologues exist with similar

domains, such as ToxB from *E. coli O157*, which shares lymphocyte inhibitory activity[8].

Lymphostatin exhibits N-terminal homology with the glycosyltransferase domain of large clostridial toxins (LCTs)[3]. It shares a cysteine protease motif with LCTs that is also found in a wider family of bacterial virulence factors[11]. LCTs play key roles in the pathogenesis of enteric clostridial infections and can drastically alter host cell morphology owing to glycosylation of factors that regulate the actin cytoskeleton[12,13]. Upon binding a surface receptor, LCTs enter the cell via clathrin-mediated endocytosis. Acidification of the endosome induces structural changes and triggers membrane insertion and translocation into the cytosol[13]. The cysteine protease domain then mediates autocatalytic cleavage of the protein to release the N-terminal glycosyltransferase (GT) domain into the cytosol, where it

[1]University of Würzburg, Rudolf Virchow Center and Biocenter, Würzburg, Germany. [2]University of Edinburgh, The Roslin Institute and Royal (Dick) School of Veterinary Studies, Edinburgh, UK. [3]Jena University Hospital, Single Molecule Microscopy Group, Jena, Germany.
✉e-mail: bettina.boettcher@uni-wuerzburg.de

subsequently inactivates host Rho GTPases that regulate multiple cellular processes, including actin polymerisation, epithelial barrier integrity, apoptosis and inflammation. The C-terminal part of LCTs is involved in receptor binding and translocation, including a delivery domain whose precise role in endosomal escape is not well understood[14]. Uptake and processing of lymphostatin has been hypothesised to occur in a similar way to LCTs[8,15] albeit only the GT domain of LifA belongs to the same protein family as the LCTs, while the delivery domain and the protease domain cluster with other families of protein toxins[16]. Substitution of a DXD motif in the predicted GT domain of LifA abolishes its activity against lymphocytes and binding of uridine diphosphate N-acetylglucosamine (UDP-GlcNAc), which has been proposed to be the sugar donor molecule[9]. Substitution of the C1480 residue in the catalytic centre of the cysteine protease domain of lymphostatin also eliminates activity, as well as the appearance of an N-terminal 140 kDa fragment in a manner dependent on endosomal acidification[15]. Thus far, only low-resolution structural analysis of lymphostatin has been performed that revealed an L-shaped molecule[9] without resolving the domain architecture.

Here, we have determined the structures of LifA by electron cryomicroscopy (cryo-EM) at different pH values, suggesting that LifA forms an inactive transport form in which the activity of the GT domain and the protease domain are blocked. This purified LifA attaches to the surface of T-lymphocytes and HEK293 cells and forms clusters that are internalised into the cells.

## Results

### LifA is a multi-domain protein in three distinct conformations

To optimise the structural preservation of LifA, we identified stabilising buffer conditions with a thermal-shift assay. LifA had the highest melting temperature ($T_M$ = 48.6 °C) at pH 6.5 in phosphate buffer, whereas neutral or slightly alkaline pH decreased the melting temperature by up to 2.1 K. The larger stabilising contribution came from the phosphate while other buffer substances at pH 6.5 lowered the melting temperature by 1–7 K (Supplementary Fig. 1). To take advantage of the stabilising effects, we used phosphate buffer throughout the purification and shifted the pH to 6.5 for the final size exclusion

chromatography, which yielded a stable protein of the expected size (Supplementary Fig. 2). Structure determination by cryo-EM and image processing showed an L-shaped LifA molecule with an N-terminal arm with abundant α-helices and a C-terminal arm with extended ß-sheets. The C-terminal arm adopted two different conformations. In conformation I the C-terminal arm was shorter and more compact, while in conformation II the C-terminal arm was extended (Supplementary Fig. 3).

We reasoned that the presence of the two conformations might be pH-dependent to trigger conformational maturation when transiting to or from the acidic endosomal compartments. To enrich a potentially acidic form, we downshifted the pH to 4.0 before vitrification. At pH 4.0, we found the same two conformations (Supplementary Fig. 3) with some 40% of LifA attributed to conformation I. The ratio between conformation I and II did not change significantly upon changing the pH and conformation II was the most abundant conformation at pH 4.0 and 6.5.

Next, we tested whether the ratio between the conformations changed towards more alkaline conditions and shifted the pH to 8.0 before vitrification. Under these conditions, we observed conformation II and another, extended conformation (conformation III, ca. 40% of LifA at pH 8.0), which was absent at the tested lower pH values. Vice versa, conformation I had disappeared, which suggested that conformation I was a low-pH form and conformation III was a high-pH form. Conformation II existed at all tested pH values, leaving it unknown whether conformation II is an intermediate when transiting from conformation I to III and back or whether it is a trapped conformation that does not respond to changes in pH.

In all three conformations of LifA, we identified six domains connected by five linkers (L–I to L–V; Fig. 1, Supplementary Fig. 4). The N-terminal arm had the GT44 domain (GT-I) (InterPro[17] PF12919) at its tip and the C58_PaToxP-like protease domain (InterPro CD20495) located near the vertex (Fig. 1a). Another GT44 domain (GT-II) (InterPro PF12919) was between these two domains and was identified with FoldSeek[18] based on its structure. The vertex of LifA was composed of an α-helical domain that had no similarity to other proteins of known structure and function. The C-terminal arm was rich in ß-sheets and

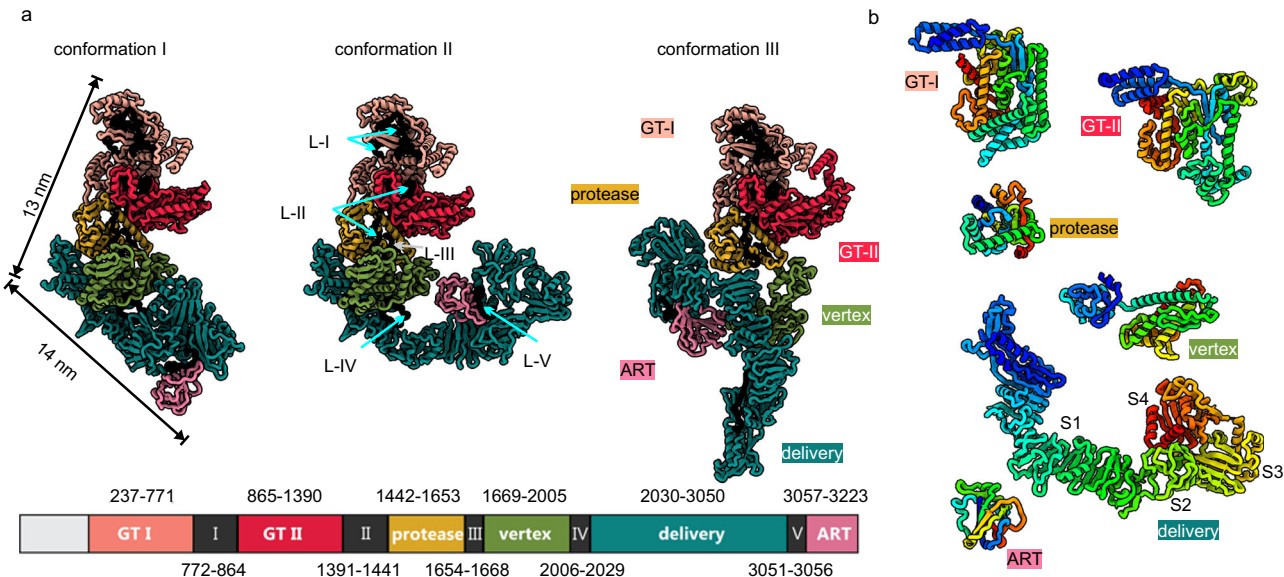

**Fig. 1 | Overall architecture of LifA. a** The models of LifA in conformation I, II (pH 4.0) and III (pH 8.0).The six domains (GT-I–light red, GT-II–red, protease–yellow, vertex–light green, delivery–dark green, and ART–pink) are shown in colour and the connecting linkers (L-I to L-V) in black. The scheme below summarises the domain organisation with the residue ranges and colour code. Additional views of the maps and models and the representations of conformation II at pH 8 are provided in the Supplementary Fig. 4. **b** The six domains of LifA are shown in an arbitrary orientation with the N-terminus close to the upper left corner. The domains are coloured in rainbow according to the residue number (N-terminus to C-terminus from blue to red). The delivery domain is subdivided into four distinct subdomains (S1–S4).

contained the domain of unknown function DUF3491 (InterPro PF11996). This domain is a characteristic of the LifA, Efa-1 and ToxB-like protein family[19] and is designated as an evolutionary conserved translocation and delivery apparatus in LCT-like toxins[20]. Therefore, we will refer to it as the delivery domain. The delivery domain was followed by an ADP-ribosyltransferase (ART) domain that was identified with FoldSeek based on its structure.

All three conformations had the same structure of the N-terminal arm with both GT domains and the protease domain while the subsequent C-terminal arm with the vertex domain, the delivery domain and the ART-domain rearranged. The rearrangement positioned the ART-domain at different locations relative to the N-terminus: In conformation I, the ART-domain was folded beneath the delivery domain. In conformation II it was close to the centre of LifA and in contact with the vertex domain and in conformation III the ART-domain was on the other side of the protease domain (Fig. 1a).

## LifA has two glycosyltransferase domains

The GT-I domain (Fig. 2a, b) is structurally closely related to the clostridial TcdA GT domain (Supplementary Fig. 5). The latter is well characterised by both X-ray crystallography and cryo-EM[21–25]. A characteristic DXD motif in TcdA coordinates a Manganese ion that stabilises the leaving phosphate group of the sugar substrate. In GT-I of LifA, this functionally essential motif[9] was found in a pocket at residues

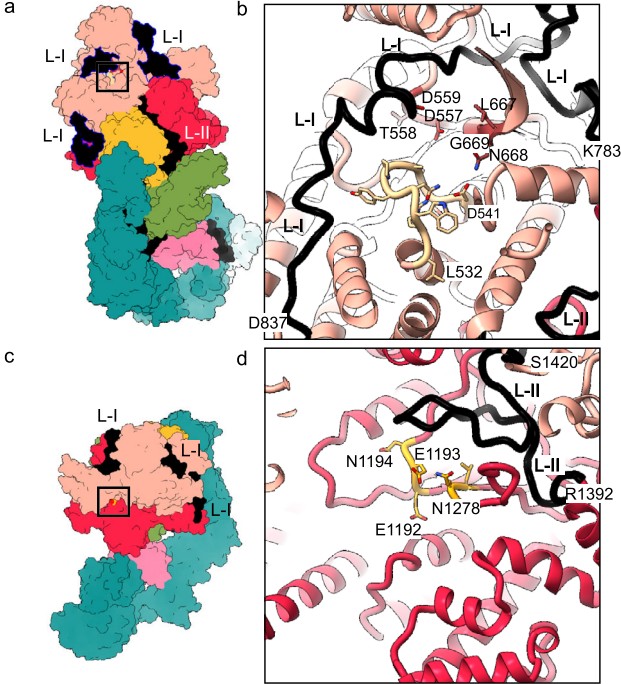

557–559 (DTD, Fig. 2b). On the opposite side of this pocket was an LNG motif (residues 667–669), which is responsible for substrate binding in TcdA and therefore marked the likely binding site of the sugar moiety of UDP-GlcNAc[9,11]. The substrate binding site of GT-I was occupied by a loop (residues 532–541, Fig. 2b). In addition, the entry to the binding cleft was blocked by the linker (L-I, residues 772–864) that connected GT-I with GT-II. L-I wrapped around the whole GT-I domain and filled the grooves on the surface of the domain (Fig. 2a). Despite the addition of Mn$^{2+}$ to the buffer, we did not observe a density close to the DXD motif that could account for a coordinated Manganese ion. This was probably due to the limited solubility of Manganese phosphate. Changing the buffer to HEPES at pH 8.0 revealed a density for a bound Manganese ion at the expected position (Supplementary Fig. 6).

The GT-II domain mapped to residues 865–1390 and was assigned as a GT domain based on its fold. FoldSeek[18] identified the GT domain of several clostridial toxins, TcdA, TcdB, TcsL (lethal Toxin from *Paraclostridium sordellii*) and TcnA (Alpha Toxin from *Clostridium novyi*), as homologues with known structure (Supplementary Fig. 5c–f). However, despite the similar fold the sequence conservation was low (13%, 14% identity), which explains why this GT domain was previously overlooked in sequence-based analyses. Notably, the GT-II domain lacked a DXD motif, instead showing an EEN motif in its place (residues 1192–1194, Fig. 2d). Although such deviations from the DXD motif are common among glycosyltransferases with a GT-A fold, we do not know whether GT-II is a fully functional glycosyltransferase. In contrast to the DXD motif in GT-I, the EEN-motif was fully exposed on the surface in a shallow indentation and was near the linker L-II (Fig. 2d).

## The active site of the LifA protease domain is inaccessible

In all three conformations, the central protease domain of LifA made van der Waals contacts and/or hydrogen bonds with the GT-II domain, the vertex domain and the delivery domain (Fig. 3a). This interaction was the same for conformation I and II, but different in conformation III, where the protease domain was embraced from several sites by the delivery domain and the ART-domain (Supplementary Fig. 7) without affecting the structure of the protease domain and of the adjacent linkers L-II and L-III.

The strictly conserved catalytic residues C1480, H1581 and D1596 constitute the active site of the LifA C58_PaToxP-like protease domain, with Q1470 stabilising reaction intermediates (Fig. 3b)[11,26]. As in the GT-I domain, the active site was shielded from the surrounding solution by a loop between the functionally relevant residues Q1470 and C1480. The loop was wedged between the active site and one of two long helices that cross each other (crossing-helices). The crossing-helices were hydrogen-bonded to the GT-II domain, the vertex domain, the functionally relevant loop of the protease, and the linkers preceding (L-II) and following the protease domain (L-III; Fig. 3c). Thus, the crossing-helices were tightly fixed in their position and could not move aside to release the functionally important loop from the entrance to the active site of the protease. This hints that the interactions of the crossing helices have a role in controlling the accessibility and activity of the protease domain.

Analogy to the LCTs suggests that the protease domain could release the N-terminal effector domains by autocatalytic cleavage[27]. A potential cleavage target within LifA is the linker L-II (residues 1391–1441) that connects the protease domain with the upstream GT-II domain. L-II was close to the active site of the protease domain but too distant for direct cleavage (Fig. 3d). It formed several hydrogen bonds with GT-I, GT-II, the vertex domain, and the crossing helices (Supplementary Fig. 8). As such, L-II glued the domains together and held LifA in a compact state. At the same time the hydrogen bonding network limited the ability of the linker to move into the active site of the protease. The cleavage of L-II would release an approximately 166 kDa

**Fig. 2 | Active sites of the glycosyl transferase domain I (GT-I) and the glycosyl transferase domain II (GT-II).** Surface representations of the model of LifA in conformation I **a**, **c**, with close-ups of the active sites of GT-I (**b**) and GT-II (**d**) in cartoon representation. The surface representations in (**a**) and (**c**) are shown in the same orientation as the active sites in (**b**) and (**d**). The black squares in (**a**) and (**c**) outline the approximate position of the close-ups in (**b**) and (**d**). Linker L-I wraps around GT-I. Note, in (**c**) the view onto the catalytic site of GT-II is occluded by GT-I. **b**, The close-up of the GT-I shows the characteristic DXD-motif for catalysis (D557, T558 and D559) and the LNG motif (L667, N668, G669) for substrate binding (Supplementary Fig. 5g). The entrance to the active site is blocked by L-I (black). The loop (residues 532–541, light yellow) blocks the substrate binding site. **d**, The close-up of the GT-II shows the EEN-motif (E1192, E1193, N1194) that superposes with the DXD motif in GT-I and in TcdA (Supplementary Fig. 5g–i). N1278 is at a similar position as N668 of the LNG motif of GT-I (L667, N668, G669). These residues are fully exposed in GT-II and are adjacent to the linker L-II (black).

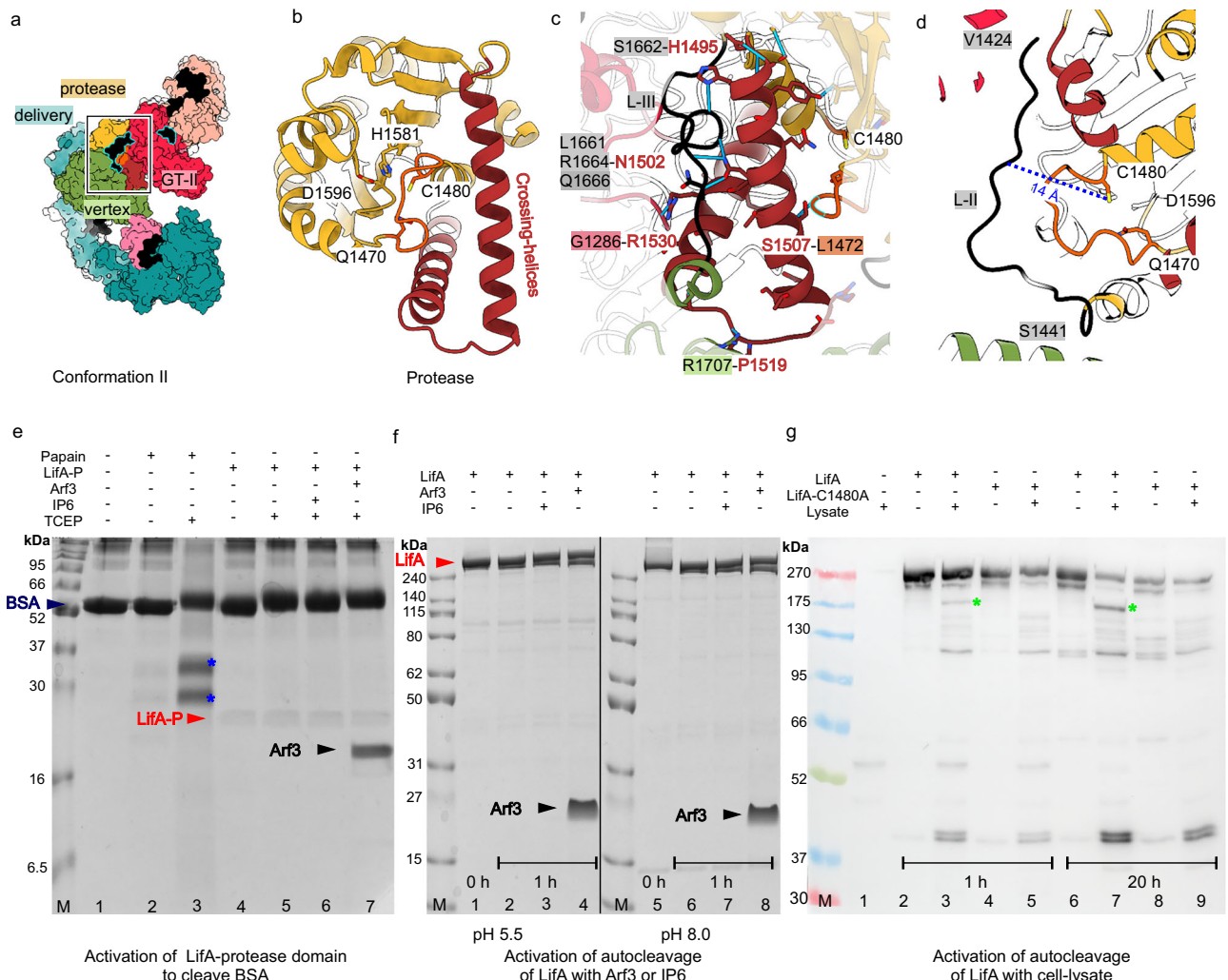

**Fig. 3 | The protease domain and its active site. a** LifA in conformation II is shown in the same orientation as the protease domain in **b**. The linker L-II is outlined in cyan. The crossing helices are shown in brown and the functionally relevant loop in orange. The protease domain is partly buried and surrounded by the glycosyl-transferase domain II (GT-II), the vertex domain and the delivery domain. The square outlines the position of the close-up in **b**. **b** Close-up of the protease domain (residues 1442–1653). The residues C1480, H1581 and D1596 form the catalytic triad and Q1470 stabilises reaction intermediates. The functionally important loop that connects N1470 and Q1480 (orange) blocks the entrance to the catalytic site. **c** Two crossing helices (brown) form a central hub that is hydrogen bonded (blue solid lines) to the vertex domain (R1707, green), the GT-II domain (G1286, red), the functionally relevant loop (L1472, orange) and the linker L-III (L1661, S1662, R1664, Q1666, grey). **d** The linker L-II (black) is the potential cleavage target and is 14 Å away from the catalytic C1480 (blue, dotted line). **e-g**, Cleavage assays of the LifA-protease domain (LifA-P) or LifA under various conditions. **e** SDS-Page of bovine serum albumin (BSA, lanes 1–7) treated with papain (lanes 2,3) or with LifA-P (lanes

4–7). Papain but not LifA-P (lanes 4–7) cleaves BSA (lane 3, cleavage products marked by blue stars) in the presence of TCEP. LifA-P is not activated to cleave BSA by the addition of IP6 (lane 6) or Arf3 (lane 7). **f** SDS-Page of LifA (lanes 1–8) incubated for 1 h at 30 °C (lanes 2–4 and 6–8) either at pH 5.5 (lanes 1–4) or at pH 8.0 (lanes 5–8). Added IP6 (lanes 3 and 7) or Arf3 (lanes 4 and 8) did not increase LifA protease activity. The complete PAGEs are shown in Supplementary Fig. 10. The experiment was repeated 2 times (*n* = 2) with similar outcome. **g** Western blot analysis of LifA (lanes 2, 3, 6, 7) or LifAC1480A (lanes 4, 5, 8, 9) treated with HEK-cell lysate (lanes 1, 3, 5, 7, 9) or with buffer (lanes 2, 4, 6, 8) after incubation for 1 h (lanes 2–5) or for 20 h (lanes 6–9) at 30 °C. The antibody was directed against the C-terminal His-Tag of LifA/LifAC1480A. Upon addition of cell-lysate, a band appeared at ca. 180 kDa (green star) in LifA but not in LifAC1480A. The experiment was repeated three times with similar outcome (*n* = 3). The blot together with the Ponceau S transfer control is shown in Supplementary Fig. 10. Source data are provided as a Source Data file.

fragment that includes both GT domains and is similar in size to the fragment found in LifA-treated bovine T lymphocytes[15].

In many toxins, the protease is activated by the binding of a cellular factor from the host cell. For LifA the activation mechanism of the protease is unknown. Inositol-hexakisphosphate (IP6) is a common host factor and activates TcdA, TcdB and RTX toxin by binding to a positively charged ß-flap[27–29] in the protease domain. Such a flap is absent in the protease domain of LifA (Supplementary Fig. 9a, b). Other toxin proteases such as the one of Makes caterpillars floppy 1 (Mcf1) and Makes caterpillars floppy-like effector-containing MARTX toxins require a host ADP-ribosylation factor (ARF) to activate auto-

catalytic cleavage[30,31]. As the LifA protease belongs to the same C58_PaToxP-like family of proteases, it is plausible that it follows a similar activation mechanism. Superposition of the proteases of LifA and Mcf1 confirmed their similarity but also showed that the N-terminal effector domains of LifA and Mcf1 have a divergent arrangement at opposite sides of the protease domain in which the ARF binding site of Mcf1 would co-locate with the N-terminal helices of the GT-II domain in LifA (Supplementary Fig. 9c, d).

We reasoned that the isolated protease domain (LifA-P, residues 1441–1653) could be active in the absence of interactions with the surrounding domains and linkers. However, LifA-P could not cleave the

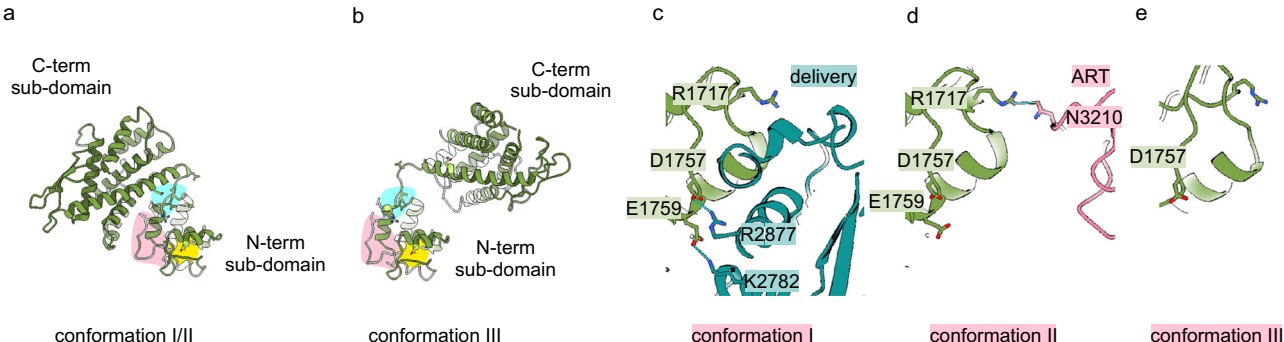

**Fig. 4 | The vertex domain interacts with different subdomains in the three conformations. a**, **b**, The vertex domain (green) consists of a smaller N-terminal and a larger C-terminal subdomain. The relative orientation of the two subdomains was different in **a** conformation I/II and **b** conformation III. The N-terminal subdomain of the vertex interacted at three sites (pink, cyan and yellow background) with different regions of LifA. In all three conformations the small subdomain was anchored to the protease domain via an H-bond between T1669 and D1524 (yellow site, Supplementary Fig. 11a–c). The N-terminal subdomain (cyan site) also bound to the N-terminal part of the S1-subdomain of the delivery domain (dark green) in conformations I/II (Supplementary Fig. 11d/e) and to the C-terminal part of the S1-subdomain in conformation III (Supplementary Fig. 11f). Another part of the N-terminal subdomain (pink background) interacted with other domains differently in all three conformations (close-ups **c–e**). **c** In conformation I it formed two salt bridges with the delivery domain (D1757–R2877 and E1759–K2782). **d** In conformation II the N-terminal subdomain formed a hydrogen bond with the ART-domain (R1717–N3210) and **e** in conformation III the N-terminal subdomain did not interact with other subdomains. **c–e** are shown in the same orientation as LifA in Fig. 1a, b) present the same viewing direction approximately perpendicular to the one in **c–e**. **a** and **b** are viewed from the GT domains towards the delivery domain.

generic target bovine serum albumin (BSA) and could not be activated under reducing conditions or the addition of IP6 or Arf3 (Fig. 3e). Next, we tested whether auto-catalytic cleavage of full length LifA could be activated by the addition of IP6 or Arf3 but observed no cleavage at pH 5.5 and at pH 8.0 (Fig. 3f, Supplementary Fig. 10). To check, whether other host cell factors could activate autocatalytic cleavage of LifA we added HEK-cell lysate and found residual proteolytic activity, probably due to endogenous proteases in the HEK-cell lysate. Western blot analysis against the C-terminal His-Tag showed the appearance of a band with an approximate molecular weight of 180 kDa, which was absent in the LifA[C1480A] mutant that lacked the cysteine in the catalytic centre of the protease (Fig. 3g, Supplementary Fig. 10). These experiments suggested that the protease of LifA follows a yet unknown activation mechanism that is independent of the known activators, IP6 and Arf3, in other toxins.

## The vertex domain provides a central anchor for stabilising different domain arrangements in the different conformations

The vertex domain had two parts (Fig. 4a,b). The smaller, N-terminal subdomain anchored the vertex domain at the connecting loop of the crossing helices of the protease domain. The larger C-terminal subdomain was a helical bundle with eight helices. It contained an aminotransferase motif (motif: TMGKALSASA, for review see[32]). However, no structural similarity of the vertex domain to transamidases was detected by FoldSeek[18]. The C-terminal subdomain adopted two different orientations relative to the N-terminal subdomain. In conformation I and II it formed a compact entity with the N-terminal subdomain, which was stabilised by a hydrogen bond between the subdomains (V1731–Q1776). In conformation III, the C-terminal subdomain was rotated by 180° reorienting it from the vertex of the L-shaped particle to a position parallel to the crossing helices of the protease domain. This rearrangement of the vertex domain left the accessibility to the catalytic sites of the upstream N-terminal effector domains unchanged.

The vertex domain was hydrogen-bonded with the adjacent protease domain, the delivery domain, the ART-domain and the linker L-III. Most of these interactions mapped to the smaller N-terminal subdomain of the vertex and targeted different sites of the LifA molecule. Some of these interactions were maintained in all three conformations, while others were specific for a certain conformation. Notably the interactions between the tip of the crossing helices at residue D1524 of

the protease domain with T1669 in the smaller subdomain of the vertex domain were maintained in all three conformations (Fig. 4a, b yellow background; Supplementary Fig. 11a–c) and anchored the vertex domain at the tip of the crossing-helices of the protease domain. The small subdomain of the vertex domain also formed a hydrogen bond with the subdomain S1 of the delivery domain close to the N-terminus of S1 in conformations I and II (V1721-R2027) and close to the C-terminus of S1 in conformation III (R1729-K2569; Fig. 4a, b cyan background; Supplementary Fig. 11d–f). The same site of the N-terminal subdomain also contained a hydrogen-bond with the C-terminal subdomain of the vertex (V1731-Q1776), which stabilised the compact inter-subdomain packing in conformations I and II and was lost in conformation III (Supplementary Fig. 11d–f). The third interaction site in the N-terminal subdomain of the vertex domain differed in all three conformations (Fig. 4a, b, pink background). This site formed two salt bridges with the S3 subdomain of the delivery domain in conformation I (Fig. 4c, D1757-R2877 and E1759-K2782)), an H-bond with the ART-domain in conformation II (Fig. 4d, R1717-N3210) and no interactions in conformation III (Fig. 4e). It is remarkable how many different interactions mapped to the small surface of the N-terminal subdomain of the vertex domain and how versatile these interactions were in the different conformations. This suggested that the N-terminal subdomain of the vertex domain provides a central hub for conformational switching and stabilisation.

## Conformations I, II and III differ by rearrangements of the delivery domain

The delivery domain was mainly composed of ß-strands and was subdivided into four ß-sandwich subdomains (S1-S4, Fig. 1b for labelling, Supplementary Fig. 12). The N-terminal ß-sandwich (S1) was the largest, containing 42 strands (residues 2030–2629), while S2-S4 had only 7–10 strands. Such ß-sandwiches are typical for bacterial translocation and pore-forming systems, as well as for receptor binding and adhesion domains[13,30,33,34]. The relative arrangement of the domains downstream of S1 (S2–S4 and ART-domain) changed between conformation I and II (Supplementary Fig. 12). In conformation I, S3 formed two salt bridges with the vertex domain (E1759-K2782 and D1757-R2877, Fig. 4c). This interaction was lost in conformation II, and instead, the ART-domain made one hydrogen bond to the vertex domain (R1717-N3210, Fig. 4d). The underlying domain rearrangement involved a rigid body rotation around G2630 of the downstream

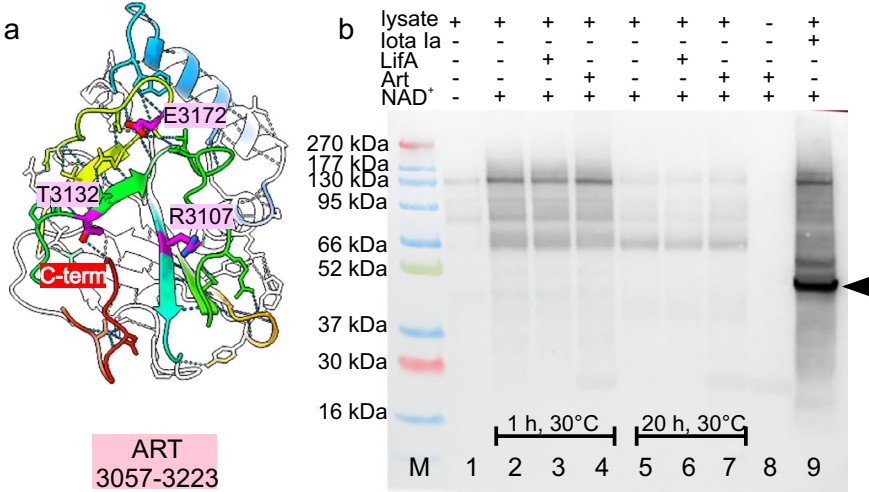

**Fig. 5 | The ART-domain has a distorted catalytic centre and has no ADP-ribosyltransferase activity on cell lysates. a** The ART-domain of conformation I (residues 3057–3223) is shown in cartoon representation (the ART-domains of all three conformations together are shown in Supplementary Fig. 14a) The residues are coloured in rainbow from the N-terminus (blue) to the C-terminus (red). The C-terminus occupies the gap between the ß-sheets and is hydrogen bonded (dotted line) with T3132. This gap is the NAD⁺ binding site in active ADP-ribosyltransferases of the R-S/T-E type. The residues R-T-E of the ART-domain are shown in magenta. These residues did not form an intact catalytic centre with R3107 pointing away from the centre. **b** Western blot analysis of ADP-ribosylation with anti-pan-ADP-ribose binding reagent (refer to Supplementary Fig. 14 for the Western blot together with the Ponceaus S loading control). HEK293T cell lysates (lanes 1–7 and 9) were incubated with NAD⁺ (lanes 2–7 and 9) in the presence of LifA (lanes 3 and 6), the ART-domain (lanes 4 and 7) or Iota Ia (lane 9). Lane 8 shows the ART-domain alone and M is the molecular weight standard. The molecular weights are indicated on the left. Samples were taken after 1 h (lanes 2–4 and 9) and after 20 h (lanes 5–7) at 30 °C. HEK-cell lysates showed NAD⁺ dependent ADP-ribosylation probably due to cellular ADP-ribosyltransferases. Addition of LifA (lanes 3 and 6) or the ART-domain (lanes 4 and 8) did not affect the ADP-ribosylation pattern. In contrast, addition of the Iota Ia, a known ADP-ribosyltransferase that acts on actin showed increased ADP-ribosylation after 1 h at 30 °C with a prominent band (black arrowhead) at the expected molecular weight of actin. The ART-assay was repeated three times independently (n = 3). Source data are provided as a Source Data file.

subdomains S2–S4 together with the ART-domain. This domain rearrangement transformed the C-terminal arm of LifA from a compact form into an extended form. The subdomain organisation of the C-terminal arm in conformation II was also maintained in conformation III (Supplementary Fig. 12). However, in conformation III the whole C-terminal arm rotated as a rigid body by 180° around linker L-IV placing the N-terminus of the S1-subdomain at the distal tip of the extended LifA. Concomitant with the reorientation of the C-terminal arm, the linker L-IV changed into an extended conformation and spanned the whole distance between the vertex domain and the distal end of LifA. In conformation III, the subdomains S2 and the C-terminus of S1 were adjacent to the protease domain. S1 formed several interactions with the N-terminal helix of the protease domain (Supplementary Fig. 13). One of these interactions was a π-cation interaction between H1448 and R2628. Such an interaction can only occur at high pH when the histidine is deprotonated and thus can contribute to the pH-dependent stabilisation of conformation III.

## The ART-domain is inactive with a disrupted catalytic centre

The ART-domain was identified as such with Foldseek based on its similarity in structure to other ART-domains. One of the hits with known structure was Vis toxin[35] (Supplementary Fig. 12f), with which it had an RMSD in the overlapping part (T3057-T3215) of the backbone of 3.7 Å and a sequence identity of 16%. Vis toxin belongs to the Clade 2 ADP-ribosyltransferases, which have a characteristic R-S/T-E motif in the catalytic centre[36]. Such an R-T-E motif was also present in the ART-domain of LifA (Fig. 5a). However, closer inspection showed that in LifA the glutamate and threonine were shifted and that the arginine side chain pointed away from the active site in all three conformations (Supplementary Fig. 14). Thus, the catalytic site of the ART-domain of LifA was disrupted. Activation would require major restructuring in which the arginine side chain would move to the other side of ß-strand. In addition to the distorted catalytic centre of the ART-domain, the

C-terminus of LifA occupied the potential binding site for the NAD⁺ substrate blocking substrate binding (Fig. 5a, Supplementary Fig. 14a).

To test whether the ART-domain of LifA was active in ADP-ribosylation, we checked for self-ribosylation by Western blot analysis. LifA, the purified ART-domain, and the ART-domain together with LifA showed no NAD⁺ dependent ribosylation in contrast to PARP3, which is known to auto-ribosylate (Supplementary Fig. 14b, c). Next, we analysed whether the ART-domain or LifA could ADP-ribosylate other proteins in a HEK-cell lysate (Fig. 5b, Supplementary Fig. 14d, e). Addition of NAD⁺ to HEK cells started unspecific ADP-ribosylation. However, the modification pattern was the same for the lysate with or without added ART-domain or LifA. In contrast, adding Iota toxin Ia (Iota Ia) generated a stronger modification signal with a prominent band with the expected molecular weight of actin (Fig. 5), which is the main target of Iota Ia[37].

## Cy3B-labelled LifA inhibits mitogen-activated proliferation of bovine T-lymphocytes and is internalised in clusters

We labelled LifA with the fluorescent dye Cy3B maleimide (LifA-Cy3B, Supplementary Fig. 15a) to study the binding and uptake of LifA-Cy3B into cells by confocal microscopy. Labelling had no impact on the structure or conformational distribution as confirmed by cryo-EM of labelled and unlabelled LifA at pH 8.0 (Supplementary Fig. 15b). Both, LifA and LifA-Cy3B, inhibited mitogen-activated proliferation of bovine T-lymphocytes with similar potency (Supplementary Fig. 15c). In bovine T-lymphocytes fixed 20 min after addition of LifA-Cy3B, fluorescent clusters started to appear in the vicinity of the plasma membrane stained with WGA633 (Fig. 6). Over time, the percentage of clusters located distal to the membrane (by at least 500 nm) increased while those proximal to the membrane decreased (Fig. 6). The formation of clusters at the cell surface followed by appearance of clusters inside the cell is consistent with uptake of LifA by receptor-mediated endocytosis.

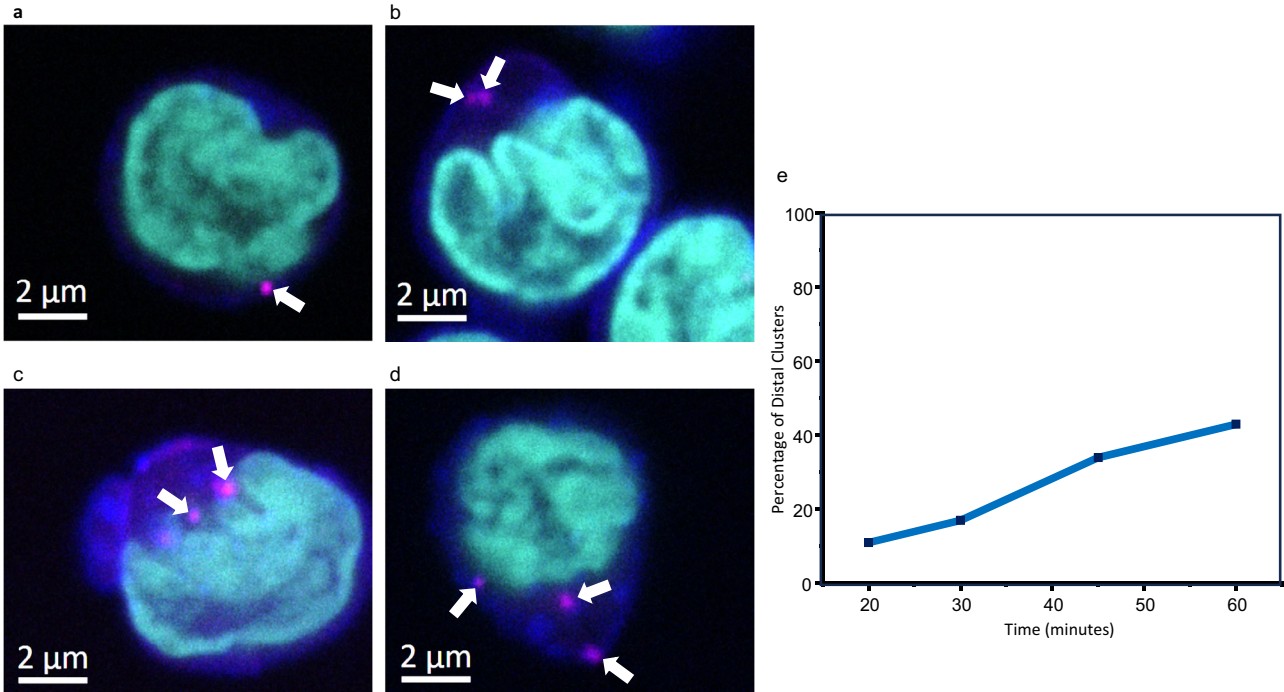

**Fig. 6 | Cy3B-labelled LifA clusters at the membrane and is internalised by bovine T-lymphocytes.** LifA formed clusters (white arrows) at the plasma membranes of bovine T-lymphocytes following incubation with Cy3B-labelled LifA. With increasing time following addition of LifA-Cy3B a greater percentage of clusters were found within T-lymphocytes distal to the cell membrane. Panels **a**–**d** show representative confocal z-section images at 20, 30, 45 and 60 min after addition of LifA-Cy3B, respectively. Cells were incubated with LifA-Cy3B (magenta) before fixing at the intervals indicated and staining with WGA633 (blue) and DAPI (cyan). For each time point at least two images were taken at a similar magnification ($n \geq 2$) as shown in panel **a**–**d**. At least eight images were taken per time point at a smaller magnification ($n \geq 8$) to show several cells per image for counting the clusters. Panel **e** shows the percentage of LifA-Cy3B clusters proximal or distal (by at least 500 nm) to the plasma membrane over time. The location of at least 100 clusters in z-stack images was analysed at each time point. Source data are provided as a Source Data file.

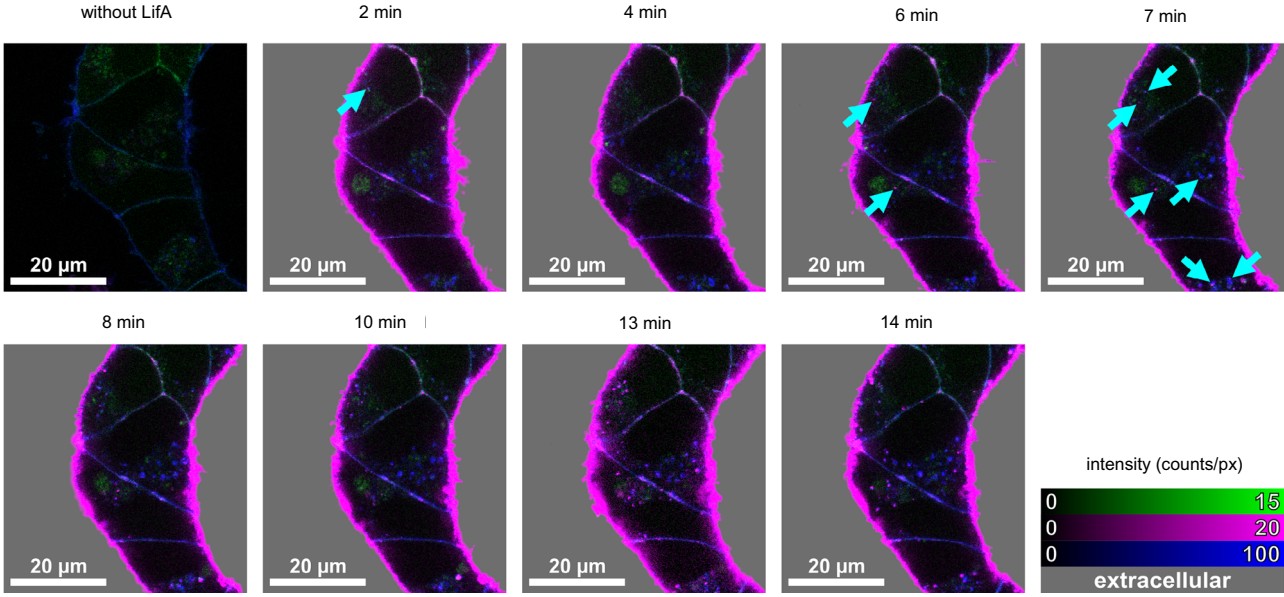

**Fig. 7 | Time-lapse imaging of LifA-Cy3B clustering on HEK-293T cell membranes and subsequent internalisation by living cells.** HEK293T cells expressing the membrane receptor NTSR1-mNeonGreen (green) were stained with WGA633 (blue). Confocal images were recorded before and after addition of LifA-Cy3B (magenta) at 37 °C. The appearance of internalised LifA-Cy3B with co-localised WGA633 spots is indicated by cyan arrows. Shown are three-colour overlay images (linear lookup tables) with shaded extracellular space (grey). The experiment was repeated three times with similar outcome ($n = 3$).

## Cy3B-labelled LifA clusters on the membranes of HEK-293T cells and is internalised

To gain further insight into the interaction between LifA-Cy3B and living cells, we used HEK-293T cells stably expressing the Neurotensin receptor 1 fused to mNeonGreen (NTSR1-mNG)[38]. These cells were exposed to LifA-Cy3B in growth medium at 37 °C. NTSR1-mNG served the purpose of highlighting the plasma membranes which was combined with WGA633 staining for three-colour confocal fluorescence lifetime imaging (FLIM) in a microfluidic chamber (Fig. 7, Supplementary Fig. 16). Immediately after addition, LifA-Cy3B started to cluster at the plasma

membranes. Internalised LifA-Cy3B clusters were found as intracellular spots within minutes. These clusters were co-localised with internalised WGA633 (Supplementary Fig. 17). The number of internalised LifA-Cy3B spots increased over time, but the average LifA-Cy3B brightness remained similar.

## Discussion

Lymphostatin is one of the largest proteins yet described in *E. coli* and possesses multiple structural domains. It has the characteristic topology of an AB exotoxin with a toxic A part consisting of the functional active domains for the effect on host cells and a structurally more variable B part involved in the host cell entry and endosomal escape. Lymphostatin follows the prototypic domain architecture of an LCT with a GT-I domain, a protease domain, and a delivery domain. Though, it also diverges from this by an additional GT-II domain and an ART-domain. Both domains are recognised by their folds but lack the characteristic motifs at their active sites.

While the GT-II domain and the GT-I domain are close together in the A part, the ART-domain is at the far side of LifA in the B part hinting at a role in a different spatial and temporal context. ART-domains are effectors in many bacterial toxins such as Cholera-like toxins, Diphtheria-like toxins, C2-like binary toxins and C3-like toxins[39]. However, the position in the B part and the combination with a GT domain in a single toxin are atypical. Most remarkable, the ART-domain changes its position relative to the effector domains of the A part in the three conformations and it contacts different sites of the vertex domain in conformation II and III. This could hint at a role in auto-processing. However, our activity assays suggest that the ART-domain is inactive, and this is also reflected by the structure that shows a disrupted catalytic centre. Thus, it is more likely that the ART-domain provides a binding and recognition platform for cellular factors than being a bona fide effector.

Based on the fold, LifA has more potential effector domains than LCTs although some are probably not functional. Other toxins also have effectors in addition to the protease and GT domain. Examples are some of the large MARTX-toxins, and the makes caterpillars floppy toxin 1 (Mcf1). MARTX-toxins have a generally different architecture, in which the effectors are sandwiched between N- and C-terminal repeats, which enable autonomous translocation across the plasma membrane of the host cell. MARTX toxins release their effectors by autocatalytic cleavage with a metal-dependent protease domain that is activated by host cell factors. However, their proteases belong to a different family of proteases than the LifA protease. Thus, the structural and functional similarities to the MARTX toxins are limited, despite a similarly large number of effector domains. More relevant is the comparison to Mcf1, which shares a PaToxP-like protease domain. The superposition of the Mcf1 protease domain[30] with the LifA protease domain shows remarkable structural similarity. However, the upstream effector domains do not overlap and have a different composition and a different arrangement relative to the protease domain. Thus, potential activation mechanisms require different interactions and different conformational responses. For Mcf1, it has been shown that the protease is activated by binding of the host protein Arf3 to a distant activator binding domain pulling the linker into the active site of the protease. Such a pulling mechanism could also be initiated in conformation III of LifA, where the delivery domain binds to the N-terminal helix of the protease domain. This helix follows the linker L-II that contains the likely cleavage target for the autocatalytic cleavage. When conformation III transforms into conformation I or II, the whole delivery domain rearranges and might pull the N-terminal helix of the protease domain away from the active site. Such movement would enlarge the catalytic cleft and move the linker L-II into the catalytic cleft for cleavage similar as suggested for Mcf1. Maintaining the interaction between the protease domain and the delivery domain, would be an important asset for transmitting the pulling force. One of

the supportive interactions is between H1448 and R2628, which is unstable in low pH environments when H1448 becomes protonated. However, changing the pH is not sufficient to activate the protease albeit an endosomal acidification is needed for C1480-dependent cleavage of LifA inside cells[15]. In addition, the protease of LifA cannot be activated by Arf3 as the one in Mcf1, instead it is triggered by a still unknown factor of cell lysates.

Bacterial toxins often adopt inactive transport forms to enable a safe delivery to the target without collateral damage at other places. Effector secretion and activation is generally tightly regulated to ensure proper progression of pathogenesis. Along these lines, we found that the GT-I domain and the protease domain are in an inactive state, with the entry to their respective substrate binding site blocked. In both cases, this blockage is due to an intra-domain loop and an extra-domain linker occluding the substrate binding pocket. Therefore, activation requires global conformational changes. These changes involve the loss of the extended hydrogen bonding network of the linkers to the surrounding domains, as well as more confined conformational changes within the domain. Conformations I-III do not show these required rearrangements. Indeed, their respective N-terminal arms with linkers L-I, L-II and L-III, both GT domains and the protease domain have the same structural arrangement in all three conformations. Thus, the reorganisation of the down-stream C-terminal arm is insufficient for the activation of GT-I and protease domain. In contrast to GT-I and the protease domain, the GT-II domain and the ART-domain lack the signature motif DXD and R-S/T-E at the expected place. Consequently, it is unlikely that these domains are functional. Instead, their importance could lie in recognition, targeting and positioning of host components to initiate the restructuring required for activation.

Lymphostatin does not induce apoptosis or necrosis in cells, but it does arrest the host cell cycle[10] and has been implicated in bacterial adherence to host cells[3,5]. It is conceivable that lymphostatin aids focal bacterial adherence, as evidenced by the formation of some large, stable clusters on the cell surface that resisted extensive washing. Immunofluorescence microscopy using antibodies against the protein found it to be enriched at the perimeter of bacterial cells in a manner like the bacterial adhesin intimin[5]. Multiple clustered LifA molecules offer interconnected interaction sites for multivalent binding with high avidity for both the host cell and the attaching bacteria. The delivery domain (DUF3491) is likely to be responsible for adherence to the host cell due to its ß-sandwich domains, which are typical for receptor binding domains. However, the primary receptor on host cells remains unknown. The cell imaging suggests that LifA adheres to the cell surface of T-lymphocytes and HEK-293T cells but does not autonomously enter the cells. Instead, the internalisation of the small clusters colocalizes with clusters of plasma membrane markers as expected for endocytosis.

It is noteworthy that while affinity-purified lymphostatin can inhibit lymphocyte function in isolation, during infection, it may also be directly injected into cells. Evidence exists that it can be secreted by a Type 3 secretion system in EPEC, the function of which is to inject bacterial proteins directly into enterocytes and is critical for intestinal colonisation[19]. Consequently, lymphostatin may interact with host cells in distinct ways, with autocatalytic cleavage following endosomal uptake and membrane insertion being one pathway, and direct injection into the cytoplasm being another. At least some of the protein also appears enriched at the bacterial surface and influences attachment[5]. Lymphostatin was the first of a wider family of lymphocyte inhibitory factors to be identified. ToxB from *E. coli O157* shares its glycosyltransferase, cysteine protease and delivery domains and is similarly able to inhibit lymphocyte proliferation and pro-inflammatory cytokine synthesis[8]. Moreover, sequencing of the genome of the EPEC strain in which lymphostatin was first described (strain E2348/69) has revealed that it contains a truncated LifA-like gene[40] which was

required for efficient formation of A/E lesions on human intestinal explants, at least when other Type 3 secreted effectors were absent[41]. Our high-resolution structures of lymphostatin have revealed a pH-dependent conformational domain rearrangement of LifA. Two of the domains were newly identified based on their fold as typical effector domains of bacterial toxins but lacked the functional signature motifs. Our structures provide rich information to interpret the potential impact of differences between lymphostatin and its homologues, both within the same strain and between pathotypes and bacterial species. It is noteworthy that lifA and lifA-like genes are encoded on integrative elements in E2348/69 and other A/E bacteria together with other effectors of the bacterial Type 3 secretion system[40]. Recombination and horizontal gene transfer events are therefore likely to have shaped its remarkable architecture and multi-functionality.

## Methods

### Protein expression and purification

The purification of LifA was adapted from[9]. In brief: Recombinant LifA containing a C-terminal His-tag in the pRham Vector (Lucigen)[9] was overexpressed in E. cloni® 10 G cells (Lucigen). All cultures contained 50 μg/ml kanamycin for bacterial selection. After overnight culture (LB medium with 0.5% (w/v) D-glucose), cells were transferred to fresh LB medium with 0.25% D-glucose and grown up to $A_{650\,nm}$ 0.8. The cells were then transferred to glucose-free TB medium containing 2 mM $MgCl_2$. At $A_{650\,nm}$ 2.0, the temperature was reduced to 30 °C for 1 h and then expression was induced with 0.2% (w/v) L-rhamnose. After 3 h, cells were harvested by centrifugation (3857 g for 27 min) and the pellet was equally distributed to four 50 ml tubes before being shock frozen into liquid nitrogen and stored at − 80 °C until further use.

For purification, one tube of shock-frozen cell pellet (containing ~ 15 g) was re-suspended in 40 ml lysis buffer which contained 20 mM sodium phosphate buffer pH 6.3, 300 mM NaCl, 500 mM 3-(1-Pyridin)-1-Propansulfonat (NDSB-201; Sigma) for protein stabilisation, 20 mM imidazole, 10% (v/v) glycerol, 100 μM phenylmethanesulfonyl fluoride (PMSF), 0.1% (v/v) Tween-20, 1 mM 2-Mercaptoethanol, one tablet of EDTA-free cOmplete™ Protease Inhibitor Cocktail (Roche; Sigma Aldrich) per 3 g of cell pellet, 2 mg DNase (Roche) and 10 mM $MgCl_2$. Cells were lysed by two passages through a benchtop cell disruptor (LM-20 Microfluidizer®; Microfluidics) at 1.25 kbar. The lysate was clarified by centrifugation (7197 g for 30 min) and applied twice at 4 °C to a Protino Ni-IDA 2000 column (Macherey-Nagel), pre-equilibrated with IMAC buffer containing 50 mM sodium phosphate buffer pH 8.0 and 300 mM NaCl. LifA was eluted in 1 ml fractions with IMAC elution buffer that contained 250 mM imidazole and was adjusted to pH 6.5. LifA was further purified by size exclusion chromatography using a Superdex 200 10/30 GL increase column (Cytiva) pre-equilibrated with SEC buffer (25 mM sodium phosphate pH 6.5, 150 mM NaCl, 75 μM $MnCl_2$, 5% (v/v) glycerol, 1.5 mM TCEP). LifA peak fractions (Supplementary Fig. 2) were pooled and concentrated to approx. 0.9 mg/ml with a 100 kDa cutoff Amicon Ultra centrifugal filter (Millipore, Merck) while glycerol was removed via washing with concentration buffer (25 mM sodium phosphate pH 6.5, 100 mM NaCl, 1 mM Uridine diphosphate N-acetylglucosamine (UDP-GlcNAc), 75 μM $MnCl_2$, 1.5 mM TCEP). The protein concentrations were determined photometrically at $A_{280\,nm}$ either with the built-in detector of the HPLC system (NGC™ Chromatography system, Biorad) or with a spectrometer (Genesys 50, Thermo Fisher). For samples to be vitrified at pH 4.0 or 8.0, the concentration buffer without UDP-GlcNAc was adjusted to pH 4.0 or pH 8.0 before use.

For the purification of the isolated protease domain (LifA-P; residues 1441–1653) or the ART-domain (residues 3065–3223) the original construct in the pRham vector was shortened in one step (ART-domain) or in two steps (LifA-P) of deletions using the Q5 site-directed mutagenesis kit (New England Biolabs). The constructs were confirmed by sequencing (Microsynth, Göttingen, Germany). The

constructs were expressed as described for the full-length LifA. Cells expressing LifA-P or the ART-domain were lysed in a buffer containing 20 mM HEPES pH 7.5, 300 mM NaCl, 500 mM NDSB-201, 20 mM imidazole, 10% glycerol, 1 mM PMSF, 0.1% Tween-20, 40 mg/L DNAse, and 10 mM $MgCl_2$ with the bench-top cell disruptor as described above. The lysate was centrifuged at 7197 g for 30 min and filtered through 0.45 μm syringe filter. The filtrate was then loaded on a 1 ml hand-packed Ni-NTA agarose column (Sigma) which was subsequently washed with 40 ml buffer containing 20 mM HEPES pH 8.0, 300 mM NaCl, 20 mM imidazole, 5% glycerol, 2 mM TCEP, 1 mM PMSF and eluted with a similar buffer containing 300 mM imidazole. For the ART-domain the buffer was exchanged with a 3 kDa spin concentrator to LifA buffer containing 50 mM HEPES pH 6.5, 150 mM NaCl, 100 μM $MnCl_2$, 5% glycerol and 1 mM TCEP. For the ART assay (see below) full-length LifA was prepared as described above but with LifA buffer for elution on the SEC. For the protease assay full-length LifA was prepared as described above but using a buffer on the SEC containing 2 mM HEPES pH 6.5, 150 mM NaCl, 5% glycerol, 1 mM TCEP and 100 μM $MnCl_2$. Furthermore, the protease inhibitor cocktail was omitted during purification.

### Protein analysis by SDS-PAGE and western blotting

Protein quality was assessed by SDS-PAGE and Western Blot. Samples were mixed with Laemmli Sample Buffer (5% ß-mercaptoethanol, 0.02% bromophenol blue, 30% glycerol, 10% SDS, 250 mM Tris at pH 6.8) and subsequently incubated at 70 °C for 10 min before being applied to the gels. The acrylamide gels were either directly stained with Coomassie (Roth) or used for Western blotting. Blotting onto nitrocellulose membranes (Cytiva) took place at 4 °C in Transfer buffer (24 mM TRIS, 194 mM Glycine, 0.05% SDS, pH 8.3) for approx. 16 h with low concentrated acrylamide gels (7.5%) to ensure complete transfer of large proteins. Membranes were blocked for 1 h with 3% (w/v) milk powder solved (Roth) in TBST buffer (150 mM NaCl, 50 mM Tris at pH 7.6, 0.1% Tween-20) before being washed with TBST. Then, membranes were incubated under soft shaking with Penta-His antibody (Penta His HRP Conjugate; Qiagen) for 1 h with 0.5% (w/v) milk powder in TBST. The final antibody dilution was 1:50,000. After washing of membranes, pre-mixed Western blot solution (1:1 mixture of peroxide solution and luminol enhancer solution, both Thermo Fisher) was applied. The membranes were imaged with a Fusion FX Imaging chamber (Vilber) with detection times of 5 to 10 min.

### Fluorescence labelling of LifA with Cy3B

For labelling of LifA with the fluorescent dye Cy3B maleimide (AAT Bioquest), washing and elution during immobilised metal affinity chromatography (IMAC) were performed at pH 8.0 to have LifA in a suitable buffer for the subsequent conjugation reaction. 400 μl of the main IMAC elution fraction was concentrated (Amicon Ultra, 100 kDa MWCO) to concentrations between 1 and 2 μM to improve labelling efficiency. TCEP concentrations were increased to between 2 and 5 μM before labelling. Labelling took place for 15 min at room temperature with 15x molar excess of dye to protein before further purification via SEC as described above (Supplementary Fig. 15a). Separation of unbound dye and the absence of oligomeric LifA-Cy3B was confirmed by fluorescence correlation spectroscopy. LifA-Cy3B was concentrated to about 1 μM as described above. For the determination of the labelling efficiency, the peak absorbance at $A_{568\,nm}$ for Cy3B was measured and related to the protein peak absorbance at $A_{280\,nm}$. The Cy3B labelling efficiency of LifA was 101%.

### ADP-ribosyltransferase assay

Synthetic constructs of the human mono-ADP-ribosyltransferase PARP3 (Uniprot: Q9Y6F1) and of Iota toxin component Ia from *Clostridium perfringens* (Uniprot: Q46220) with optimised sequences for expression in *E. coli* were obtained from Geneart (Thermo Fisher,

Regensburg, Germany) and subcloned into the first cloning site of the pET-Duet-1 vector (Novagen). DNA fragments with overlapping regions were obtained by PCR and inserted with the Hifi assembly kit (NEB) in frame with a sequence coding for a N-terminal His$_6$-tag. The correctness of the resulting constructs, pET_hPARP3 and pET_Iota-Ia, were confirmed by sequencing (Microsynth, Göttingen, Germany). The constructs were transformed into BL21(DE3) and proteins were expressed by induction with 1 mM IPTG for 4 h at 30 °C in LB medium. Cells were harvested by centrifugation at 5000 g and stored at −80 °C until further use.

Cells expressing hPARP3 were lysed by re-suspension of the cell pellet in a buffer containing 50 mM Tris HCl pH 8.0, 300 mM NaCl, 10% glycerol, 5 mM MgCl$_2$, EDTA-free protease inhibitor cocktail (Roche), 1 mM PMSF, and 1.6 mg/ml lysozyme and sonication. The lysate was centrifuged for 30 min at 7200 g and 4 °C. The supernatant was filtered through a 0.45 μm syringe filter and loaded on 1 ml hand-packed Ni-NTA agarose column (Sigma). The column was washed with 40 ml IMAC buffer containing 50 mM Tris HCl 8.0, 300 mM NaCl, 20 mM imidazole, 10% glycerol, and 10 mM MgCl$_2$ and eluted with the IMAC buffer containing 300 mM imidazole. The peak fractions were subjected to size exclusion chromatography using a 10/300 Superdex75 column (Cytiva) at 0.75 ml/min with a buffer containing 20 mM HEPES pH 7.5, 100 mM NaCl and 5 mM MgCl$_2$.

Cells expressing Iota Ia were resuspended in a lysis buffer containing 50 mM sodium phosphate buffer pH 7.5, 300 mM NaCl, 5 mM EDTA, 1.6 mg/ml lysozyme, 1 mM PMSF, a protease inhibitor cocktail (Roche), and 10% glycerol and lysed by sonication. 0.1 mg/ml DNase (Roche) and 10 mM MgCl$_2$ was added and incubated for 30 min on ice. The lysate was then centrifuged at 7200 g for 30 min at 4 °C and the supernatant was filtered through a 0.45 μm syringe filter. The filtrate was loaded on a 0.5 ml hand-packed Ni-NTA agarose (Sigma) column and washed with 20 ml washing buffer, containing 50 mM sodium phosphate buffer pH 7.5, 300 mM NaCl, 10% glycerol and 20 mM imidazole. Elution followed with the elution buffer (same as washing buffer but containing 300 mM imidazole) and the fraction containing Iota Ia was concentrated with a 30 kDa spin concentrator (Millipore).

A HEK cell lysate was prepared by resuspending 1*10$^7$ cells in 0.5 ml lysis buffer containing 50 mM Tris HCl pH 8.0, 150 mM NaCl, 1% Triton X-100 and protease inhibitor cocktail (Roche). The cells were repeatedly mixed by pipetting during an incubation on ice for 30 min. After centrifugation at 13,000 g for 10 min at 4 °C, aliquots of the cell lysate were frozen at −80 °C until further use.

To test if LifA (0.3 μM final concentration) or the isolated ART-domain (16.2 μM final concentration) modify themselves with ADP ribose, the purified proteins were incubated for 1 h at 30 °C with or without 250 μM NAD$^+$ in a total volume of 30 μl. As positive control 1.4 μM hPARP3 was used. The ART reaction buffer contained 50 mM HEPES pH 6.5, 150 mM NaCl, 6.7 mM MgCl$_2$, 100 μM MnCl$_2$, 5% glycerol and 1 mM TCEP. The reaction was stopped by addition of 10 μl SDS sample buffer and 10 min incubation at 70 °C. SDS-PAGE electrophoreses was performed with 4–12% gradient gels (Invitrogen) with MES running buffer (Invitrogen). Proteins were blotted for 2 h at 10 V onto a nitrocellulose membrane. After Ponceau S staining as loading control, membranes were blocked with 3% milk powder suspension (Roth) in TBST buffer (50 mM Tris pH 7.6, 150 mM NaCl, 0.1% Tween-20) for 1 h at room temperature. The membrane was then incubated with 1:3000 Anti-pan-ADP-ribose binding reagent (MABE1016, Millipore) in 0.5% milk powder suspension for 1 h at room temperature. The membrane was washed with TBST buffer and incubated with 1:10,000 Rabbit IgG secondary antibody HRP conjugate (31460, Thermo Fisher) in 3% milk powder suspension for 1 h at room temperature. ECL Western blotting substrate (32,106, Thermo Fisher) was used for detection. Two repeats of the assay were performed ($n$ = 2).

ADP ribosyltransfer on cell targets were tested with 5 μl HEK cell lysate incubated with LifA (0.3 μM final concentration) or the isolated

ART-domain (9.5 μM final concentration) and 25 μM NAD$^+$ for 1 h or 20 h in a total volume of 30 μl in ART reaction buffer. As positive control, the ADP ribosylation of actin by 61 nM Iota Ia toxin was used under the same conditions. SDS-PAGE, Western blotting and detection were identical to the other ART assay described above. Three repeats of the assay were performed ($n$ = 3).

### Protease activity assay
The point mutation C1480A in LifA was introduced using the Q5 site-directed mutagenesis kit (New England Biolabs) and confirmed by sequencing (Microsynth, Göttingen, Germany).

A synthetic construct of the human ADP-ribosylation factor 3 (Arf3; Uniprot: P61204) with an N-terminal deletion of 17 amino acids and a Q71L mutation was obtained from Geneart (Thermo Fisher, Regensburg, Germany), which was sequence-optimised for expression in *E. coli*. The gene was subcloned into a pTrc99A-His10 vector. A DNA fragment with overlapping regions was obtained by PCR and inserted with the Hifi assembly kit (New England Biolabs) in frame with a sequence coding for a C-terminal His10-tag. The correctness of the resulting construct pTrc_hArf3 was confirmed by sequencing (Microsynth, Göttingen, Germany). The construct was transformed into BL21(DE3) and Arf3 was expressed by induction with 1 mM IPTG for 4 h at 30 °C in LB medium. Cells were harvested by centrifugation at 5000 g and stored at −80 °C until further use.

Cells expressing hArf3 were lysed by re-suspension of the cell pellet in a buffer containing 50 mM Tris HCl pH 8.0, 300 mM NaCl, 1% Triton X-100, 10% glycerol, 5 mM MgCl$_2$, EDTA-free protease inhibitor cocktail (Roche), 1 mM PMSF, and 1.6 mg/ml lysozyme and incubation on ice for 1 h. The lysate was centrifuged for 30 min at 7200 g and 4 °C. The supernatant was filtered through a 0.45 μm syringe filter and loaded on 1 ml Ni-IDA agarose column (Protino Ni-IDA 2000, Machery-Nagel). The column was washed with 40 ml IMAC buffer and eluted with the IMAC buffer containing 300 mM imidazole. The peak fractions were subjected to size exclusion chromatography using a 10/300 Superdex75 column (Cytiva) at 0.75 ml/min with a buffer containing 20 mM HEPES pH 7.5, 100 mM NaCl and 5 mM MgCl$_2$. Fractions containing Arf3 were concentrated with 3 kDa spin concentrators (Millipore).

HEK cell lysates were made the same way as for the ART assays but with 1 mM PMSF instead of the protease inhibitor cocktail in the lysis buffer.

Protease assays to test the dependence on cellular factors were performed with or without 5 μl HEK cell lysate at 30 °C for 1 h or 20 h in a total volume of 31 μl. 20 mM succinic acid, 70 mM sodium phosphate and 70 mM glycine at a pH 8.0 (SPG 8.0) was used as buffer and was supplemented with 5.8 mM TCEP. 20 μl LifA WT (final concentration 0.25 μM) or LifA C1480A mutant (final concentration 0.35 μM) were added to the reaction in a buffer containing 2 mM HEPES pH 7.5, 150 mM NaCl, 100 μM MnCl$_2$, 5% glycerol and 1 mM TCEP. The reaction was stopped by the addition of 10 μl SDS sample buffer and 10 min incubation at 70 °C. SDS-PAGE was performed with 8% hand-cast gels and Tris-glycine buffer or 4–12% gradient gels (Invitrogen) and MES buffer (Invitrogen), followed by 2 h blotting at 10 V on nitrocellulose. After Ponceau S staining as loading control, membranes were blocked with 3% milk powder suspension (Roth) in TBST buffer for 1 h at room temperature. The membrane was then incubated with 1:10,000 Anti-Penta His HRP conjugate (1014992, Qiagen) in 0.5% milk powder suspension for 1 h at room temperature. ECL Western blotting substrate (32,106, Thermo Fisher) was used for detection. Three repeats of this assay were performed ($n$ = 3).

Protease assays were also performed with known activators of other toxins: LifA WT (final concentration: 0.28 μM) was incubated with 0.9 mM D-myo-inositol-hexakisphosphate (4,07,125, Sigma) or 3.2 μM Arf3 together with 0.9 mM GTP (Roth), 6 mM MgCl$_2$ for 1 h or 20 h at 30 °C in a total volume of 33 μl. Controls without these activators were

also prepared. The pH was adjusted to pH 5.5 or pH 8.0 with SPG 5.5 or SPG 8.0 and 4.5 mM TCEP was added. A control without TCEP and without incubation was also taken. The reaction was stopped by the addition of 10 µl SDS sample buffer and 10 min incubation at 70 °C. SDS-PAGE electrophoreses were performed with 4–12% gradient gels (Invitrogen) with MES running buffer (Invitrogen) and afterwards stained with Coomassie R-250. Two repeats of this assay were performed ($n = 2$).

### Vitrification of samples and EM data acquisition

Grids (UltrAuFoil® R 0.6/1 or holey carbon foil R 1.3/1,3 300 mesh; Quantifol) were glow discharged in air at a pressure of $3.0 \times 10^{-1}$ Torr at medium power for 120 s–150 s with a Harrick Plasma Cleaner (PDC-002). The grids were plunge frozen in liquid ethane using a Vitrobot IV (FEI). The settings were 4 °C, 95% humidity, blot time 5 s, blot force of 25, without drain and wait time.

Movies were acquired with a Titan Krios G3 (Thermo Fisher) at 300 kV using EPU. For lymphostatin at pH 6.5, the Falcon III direct detector (Thermo Fisher) was used in linear mode at a nominal magnification of 75.000 and a total exposure of 73 e-/Å$^2$ (Supplementary Table 1). At each stage position, movies were obtained from the central hole and the four closest surrounding holes using image shift without beam tilt compensation. Movies were stored in MRC-format and motion corrected and dose weighted with MotionCor2[42] during the data acquisition.

For lymphostatin at pH 4.0 and pH 8.0, movies were acquired at the same microscope with a Falcon IVi direct detector and a Selectris energy filter (Thermo Fisher) in counting mode with zero-loss imaging (slit width of 5 eV). Movies were obtained with the fast option of EPU that uses aberration-free image shift (AFIS)[43] within a radius of 12 µm of the stage position. The holes were selected based on the plasmon image of a grid mesh[44] and the ice filter was adjusted for a narrow distribution of the ice thickness. The movies were acquired in EER format and pre-processed in a CryoSPARC Live session[45] including patch motion correction, dose weighting, patch CTF determination, blob picking, particle extraction and 2D-classification (Supplementary Table 1).

### EM image processing and model building

The image processing and the initial ab-initio reconstructions were done with CryoSPARC (versions 4.0–4.6)[45] using standard procedures. The particle images had intrinsic flexibility, preferential orientation and two conformations. Therefore, intense sorting with several rounds of 2D-classification, and 3D-classification with heterogeneous refinement or variability analysis were required to identify homogeneous subsets of the particles. Individual clusters and/or classes were refined with nonuniform refinement. With this strategy, subsets of particles were identified, which were more isotropic in resolution than others and helped to identify subsets belonging to conformations I-III. Within a subset, the resolution of the flexible parts could not be significantly improved with Flex-EM. Therefore, the respective subset was transferred to RELION 5.0[46] using pyem[47]. In RELION, the particles were further classified and refined to generate a consensus refinement of the full particle and focused refinements of the individual domains. The workflows for the subsets are summarised in Supplementary Figs. 18–26. Improving the resolution of flexible parts in the consensus maps of conformations I-III with flex EM in CryoSPARC did not show a significant improvement, and we could obtain more complete maps with the strategy described above.

The interpretability of the maps was judged by automated model building with ModelAngelo[48] based on the completeness of the models. The local resolution of the maps and the angular distribution of the particles was determined with RELION and is shown in Supplementary Fig. 27.

### Model building.

Composite maps were generated using the consensus map of the full particle and the maps of the focussed refinements with the Combine Focused Maps tool in the programme Phenix[49]. For conformation I, manual model building with Coot[50] was guided by automated model building via ModelAngelo[48]. Model building of conformation II was started with the model of conformation I. Conformation III was started with the model of conformation II fitting the C-terminal arm as a rigid body. Real-space refinement was done with Phenix[49]. The final models are refined against the composite maps (Supplementary Tables 2–7). For further evaluation, selected helices of the models are shown together with the respective composite maps in Supplementary Fig. 28.

### Domain and linker assignment.

Initial assignment of the GT-I domain, the protease domain and the DUF3491 followed the assignment in the INTERPRO database[17]. The GT-II domain, and the ART-domain were identified with Foldseek[18]. The ribfind[51] plugin in ChimeraX[52] was used to identify the linkers as regions that did not cluster within a designated rigid body and connecting the designated domains.

### Bovine T-lymphocyte proliferation assays and confocal microscopy

Bovine venous blood was collected in accordance with the Animals (Scientific Procedures) Act 1986 with consent from the local Animal Welfare & Ethical Review Board. Briefly, 50 mL blood was collected from 12 to 18-month-old Holstein-Friesian cows using citrate phosphate dextrose-adenine (CPDA-1) syringes per animal per sampling day. Isolation of peripheral blood mononuclear cells and subsequent enrichment of T cells using nylon wool columns was performed[9]. Enriched T cells were used to assess the ability of wild-type LifA and Cy3B-labelled LifA to inhibit mitogen-activated proliferation[9]. Briefly, LifA and LifA-Cy3B were added at the concentrations indicated in Supplementary Fig. 13. Proliferation was stimulated using concanavalin A (ConA; Sigma) at 1 µg/mL final concentration. Cells were incubated at 37 °C for 72 h. CellTiter 96® Aqueous One (Promega) was added 18 h before the assay end. Absorbance measurements were taken using a MultiSkan Ascent plate reader (Thermo Scientific) at 490 nm. Untreated cells were used as negative control, and background medium value subtracted from all values. All conditions are represented as a proliferation index (Absorbance of cells treated with ConA and LifA/Absorbance of cells treated with ConA alone).

For confocal microscopy, $4 \times 10^6$ enriched T-lymphocytes were plated into one well of a 24 well flat bottom plate. LifA-Cy3B was added at a final concentration of 60 nM. Cells were incubated at 37 °C, 5% CO$_2$ for 60 min. Cells were removed at 2, 4, 6, 8, 10, 15, 20, 30, 45 and 60 min post LifA-Cy3B treatment and fixed in 4% (w/v) paraformaldehyde in PBS. Cells were washed with dH$_2$O, blocked with 2% (w/v) bovine serum albumin in PBS for 1 h, and stained with Wheat Germ Agglutinin Alexa Fluor 633 conjugate (W21404, Thermo Scientific) and 4′,6-diamidino-2-phenylindole (DAPI; D3571, Thermo Scientific) as directed by the manufacturer. Cells were again washed, then resuspended in 10 µL dH$_2$O, pipetted onto microscope slides, air dried and ProLong diamond antifade mounting medium added (p36961, Invitrogen). For confocal imaging, an LSM880 microscope was used with a 63x oil immersion objective for z-stack imaging.

### Time-lapse fluorescence imaging of living HEK293T cells

HEK293T FlipIn cells (Thermo Scientific) stably expressing the G protein-coupled receptor neurotensin receptor 1 linked to the fluorescent protein mNeonGreen, NTSR1-mNeonGreen[38,53], were seeded into microfluidic chambers with 170 µm cover glass bottom (µ-Slide VI$^{0.5}$, IBIDI) 3 days in advance. The cells grew at 37 °C to about 30% confluency in growth medium comprising DMEM (ThermoFischer Scientific) supplemented with 10% foetal bovine serum (FBS)

(Bio&Sell), 1% penicillin/streptomycin (Sigma) and, 3 μg/ml puromycin (InvivoGen). The plasma membranes were stained with WGA633 (Wheat Germ Agglutinin Alexa Fluor 633 conjugate, Thermo Scientific). for 15 min by addition of 1μl of the marker solution (1 mg/ml) to the growth medium inside the microfluidic chamber. Next, 6.67 μl LifA-Cy3B (0.9 μM in HEPES buffer, i.e., ~60 nM final concentration) was added in one entry port of the microfluidic chamber on the microscope without replacing the growth medium including WGA633, to record the same cells before and after the addition of LifA-Cy3B. Cells were kept at 37 °C by a stage top incubator system (OKO lab) during the microscopic measurements. The experiment was repeated 3 times ($n = 3$).

Three-colour fluorescence imaging microscopy was recorded with a confocal STED-FLIM microscope (ExpertLine, Abberior Instruments) using the software Imspector (Abberiro Instruments). Picosecond pulsed laser excitation was applied with 488 nm for mNeonGreen, 561 nm for Cy3B and 640 nm for WGA633 at 40 MHz repetition rate. A 60x water immersion objective with numerical aperture 1.2 (UPLSAPO 60XW, Olympus) allowed extended z-stack imaging. Fluorescence of mNeon-Green was detected in the spectral range from 500 to 550 nm. Fluorescence of Cy3B was detected in the spectral range from 580 to 630 nm, and fluorescence from WGA633 was detected in the spectral range from 650 to 720 nm. Each spectral range was separated by a polarising beam splitter cube to enable fluorescence anisotropy analysis. Photons were recorded by six single-photon counting avalanche photodiodes (APDs) with optimised time resolution (SPCM-AQRH-14-TR, Excelitas) using synchronised TCSPC electronics (SPC-154N, Becker & Hickl). Signals from different detectors could be combined if needed. Time binning was set to 21 ps to obtain high-resolution fluorescence lifetime decays. Images were not processed beyond changing the maximum intensity of the display look-up tables.

**Reporting summary**

Further information on research design is available in the Nature Portfolio Reporting Summary linked to this article.

## Data availability

The EM-maps generated in this study have been deposited in the Electron Microscopy Data Bank (EMDB) under accession codes: EMD-19982 (conformation I, pH 4.0, consensus), EMD-19983 (conformation I, pH 4.0, focused C-terminal part), EMD 19987 (conformation I, pH 4.0, composite map), EMD-19984 (conformation II, pH 4.0, consensus), EMD-19985 (conformation II, pH 4.0, focused C-terminal part), EMD-19988 (conformation II, pH 4.0, composite map), EMD-53286 (conformation I, pH 6.5, phosphate buffer), EMD-53287 (conformation II, pH 6.5, phosphate buffer), EMD-53168 (conformation II, pH 8.0, phosphate-buffer, consensus map), EMD-53167 (conformation II, pH 8.0, phosphate-buffer, focussed on N-terminal arm), EMD-53166 (conformation II, pH 8.0, phosphate-buffer, focussed on centre), EMD-53165 (conformation II, pH 8.0, phosphate-buffer, focussed on S1-subdomain), EMD-53164 (conformation II, pH 8.0, phosphate-buffer, focussed on C-terminal arm), EMD-53169 (conformation II, pH 8.0, phosphate-buffer, composite map), EMD-52985 (conformation II, pH 8.0, HEPES-buffer, consensus map), EMD-52986 (conformation II, pH 8.0, HEPES-buffer, focussed on N-terminal arm), EMD-52987 (conformation II, pH 8.0, HEPES-buffer, focussed on centre), EMD-52988 (conformation II, pH 8.0, HEPES-buffer, focussed on S1-subdomain), EMD-52989 (conformation II, pH 8.0, HEPES-buffer, focussed on C-terminal arm), EMD-52990 (conformation II, pH 8.0, HEPES-buffer, composite map), EMD-52995 (conformation III, pH 8.0, consensus map), EMD-52994 (conformation III, pH 8.0, focussed on N-terminal arm), EMD-52992 (conformation III, pH 8.0, focussed on centre), EMD-52991 (conformation III, pH 8.0, focussed on S1-subdomain), EMD-52993 (conformation III, pH 8.0, focussed on C-terminal arm) and EMD-52996 (conformation III, pH 8.0, composite map).

The molecular models generated in this study have been deposited in the Worldwide Protein Data Bank (wwPDB) under accession codes: 9EUV (conformation I, pH 4.0), 9EUW (conformation II, pH 4.0), 9QHH (conformation II, pH 8.0, phosphate buffer), 9QB (conformation II, pH 8.0, HEPES buffer) and 9QBB (conformation III, pH 8.0). The raw data generated during the current study are available from the corresponding author on request. Uncropped gels, and Western blots can be found in the Source data. Source data are provided with this paper.

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

## Acknowledgements

Cryo-electron microscopy was carried out in the cryo-EM facility of the Julius-Maximilians-Universität Würzburg funded by the Deutsche Forschungsgemeinschaft (DFG, German Research Foundation – Projects INST 93/903-1 #359471283, INST 93/1042-1 #456578072, INST 93/1143-1 # 525040890). B.B. acknowledges project funding by the DFG (Project Bo1150/18-1 #428774170). M.P.S. acknowledges strategic investment by the Biotechnology & Biological Sciences Research Council (BBS/E/RL/230002 C). M.B. acknowledges funding by the State of Thuringia and the DFG for the confocal FLIM-STED microscope, Abberior Instruments (DFG project INST 1757/25-1 #411346541).

## Author contributions

M.G., T.R., M.B., L.S. and B.B. developed the experimental design. M.G. and R.S. purified LifA and labelled the protein. M.P.S. and M.H. did T-lymphocyte proliferation and uptake assays. T.R. and R.S. did the protease and the adenylation assays, including expression and purification of all proteins. C.K., T.R. and B.B. collected the EM-data. B.B. processed the electron microscopic image data, L.S., M.H., M.G., M.P.S. and M.B. performed the light microscopic experiments. L.S., M.H., M.P.S. and M.B. processed the light microscopic data. T.R., M.G. and B.B. built the molecular models. V.J.F., M.G., L.S., M.B. and B.B. designed the figures. M.G., T.R., M.P.S., M.B. and B.B. interpreted the data. All authors contributed to the writing and editing of the manuscript.

## Funding

## Competing interests

The authors declare no competing interests.
