## [Transparent Peer Review file · Nature Communications]

Structure of lymphostatin, a large multi-functional virulence factor of pathogenic *Escherichia coli*

Corresponding Author: Professor Bettina Böttcher

Version 0:

Reviewer comments:

Reviewer #1

(Remarks to the Author)

Griessmann and colleagues describe the structure of lymphostatin from O127:H6 enteropathogenic *Escherichia coli* using single particle cryo-electron microscopy. Lymphostatin (365 kDa) is reported to be an inhibitor of mitogen and antigen activated proliferation of lymphocytes and proinflammatory responses. It has also been reported to mediate bacterial adherence to cells and influences actin nucleation at sites of attachment. Its glycosyltransferase and cysteine protease motifs are required for activity on lymphocytes, but structural information has remained unknown prior to this manuscript.

The structural analysis reveals a multi-domain protein in two distinct conformational states. The domains are connected by long linkers that in some cases occlude access to the active sites. The structures are exciting in that they also reveal the unexpected presence of an ADP-ribosyltransferase (ADPRT) domain fold at the C-terminus and a second glycosyltransferase (GT) domain fold. The structures appear to be of high quality and are expected to be a major advance for the toxin and *E. coli* pathogenesis fields.

While the structure is a significant accomplishment, the manuscript falls short of what one might expect for a high impact publication. The first weakness is in the significant number of questions that are left unanswered. While experimentally answering some of these may arguably fall outside of the scope of this manuscript, others are simple to address and at least merit discussion. The second pertains to the organization; it was hard to appreciate the structural details as key points were split across multiple figures.

Major points and unanswered questions.

1. The authors make a point to say that LifA can adopt two different conformations and that this conformational flexibility does not depend on pH. They only test pH 4.0 and 6.5 though. What happens when the pH is above neutral, eg. pH 8.0?
2. The authors make a point to say that the cysteine protease activity depends on low pH. Why? In the case of the LCTs, zinc is needed to remove an inhibitory zinc. Is a zinc observed in the cysteine protease active site?
3. The analogy to the pro-domain of a caspase is not well supported. The structural overlap was not clear from the figures. Further, is there any experimental evidence to suggest that this is a mechanism of activation in LifA? The more obvious hypothesis based on the homology to the PaTox/LCT cysteine protease domains is that there is an allosteric activator such as InsP6. Is there any structural evidence to suggest that an allosteric activator could bind another portion of this domain?
4. The authors make a point to say that the active sites of GT-I and the cysteine protease domain are occluded. Are the sites equally occluded in both structural states? If not pH, can they speculate on what will be needed to induce an accessible and active enzyme?
5. The authors report a C-terminal ADPRT fold. Are the key conserved residues for ADPRT activity present? Does the domain bind NAD? This is easy to test.
6. Lines 98 and 108. Given the strong structural homology between the TcdA and TcdB GT domains, it is potentially misleading to imply a distinction where GT-I is more like TcdA GT and GT-II is more like TcdB GT. Is there a biological reason to make this distinction?

Organizational points:

In many places, it would help to have simpler, larger, and higher resolution figures. (For example, Figure 1 is great.) I recommend that the authors present their figures in the order that they present the text describing the results. Please explain the content in the extended data.

Line 67 – Extended Data Figure 1 statement is unclear what it is referring to in the panel. Likely panels F & G. If so, move them so they're panels A & B so the figure is in the order in which they are referred to in the manuscript. Explain the other panels.

Lines 102 – 108 – The text description does not match the figure. There are no annotated residues for occupied loops (residues 532-542, figure 2a) or to the referred linker L-1 into the binding cleft.

Line 129 – Coming back to figure 2 is confusing. I would have Figure 2 focused on the glycosyltransferase fold and devote Figure 3 to the protease site. This would allow the authors to modify the close-up orientations in Figures 3b-e to allow the reader to understand where the described residues are located. Currently, the close-ups differ from the overview.

The rationale for transitioning to Figure 4 was not obvious and did not add much in my view.

Other points:

Lines 25-27: It is misleading to say that LifA has more functional domains than any other bacterial toxin. No evidence to suggest GT-II and ADPRT domain are functional and RTX toxins can have five or more enzyme activities.

Line 44. References 8/9 do not seem appropriate for this statement.

Extended data Figure 2. It would be helpful to know how panel a relates to the orientations shown in b.

Line 98 – Why skip to “Extended Figure 4a (sic)” when Extended Data Figure 3 has not been described?

Extended Data Figure 3 is not talked about in the text. While nice to have, it would be helpful if panels were larger with less white space. Same with Extended Figure 4. Extended data figure 5 is great.

Figure 4 and Extended data Figure 6. Recommend changing colors to accommodate red/green colorblind individuals. Gray scale for individual channels is easiest to see.

Minor points:

Line 101 – Add comma after LifA

Reviewer #2

(Remarks to the Author)

Lymphostatin is a virulence factor of pathogenic *E. coli*, whose molecular mechanism of action remains largely unclear. In this manuscript, the group of Bettina Böttcher determined two high-resolution structures of this large bacterial toxin and described its architecture at the molecular level. The results obtained are original and significant for the field. The manuscript contains structural data of outstanding quality, from which the authors draw far-reaching conclusions about the functionality of the toxin without any additional biochemical studies. The latter issue needs to be addressed before publication.

1. The authors state that lymphostatin has the highest number of domains of any known toxin. I am surprised by this claim since Multifunctional-autoprocessing repeats-in-toxin (MARTX) toxins typically have 5+ effectors and several other domains necessary for efficient delivery and function. For review: <https://doi.org/10.1128/microbiolspec.ve-0002-2014>.
2. Line 82: The authors describe the identification of an ADP-ribosyltransferase domain at the C-terminus of the toxin. This is a very interesting discovery. But what is the evidence that this domain has this particular function? The structure itself is not sufficient for such a statement. Please provide the results of functional experiments.
3. Line 108. Is the structural data the only evidence that GT-2 is a glycosyltransferase? I am skeptical of this claim, especially because this domain lacks a common DXD motif. Please perform functional studies to show that the GT-2 domain performs this function.
4. Line 127. The unavailability of the protease domain is also asserted based on structure alone. TcdA and TcdB toxins are activated by inositol phosphate cofactors <https://www.nature.com/articles/nature05622> and Mcf-like toxins are activated by

Arf proteins <https://www.pnas.org/doi/abs/10.1073/pnas.1905095116> I suggest that the authors test whether the addition of these molecules activates protease activity.

5. If both conformations I and II are present at pH 4 and 6.5, what was the rationale for determining their structures at both pHs? Are they physiological states or could one of them be an artifact in protein production? I suggest that the authors measure the LD50 of the toxin on cell cultures and compare it with previous reports.

6. Line 185. I am confused by the last statement of this paragraph. Bacterial toxins are known to act quickly and efficiently, for example, *C. difficile* toxins A and B are internalized within 20-30 minutes. If the toxin is functional, it should not take 15 hours for it to penetrate target cells.

7. Line 243. If the data suggest T3SS-mediated toxin entry, why does the toxin need a delivery domain? Please discuss.

8. The protein expression protocol has several interesting elements that are not adequately discussed.

8.1. Why did you purify the protein at pH 6.5 in the first place?

8.2. (1-Pyridine)-1-Propanesulfonate was added to stabilize the protein. How did it help? Could it have affected the structure of the protein, leading to the appearance of two conformations?

8.3 Line 288-289. Uridine diphosphate N-acetylglucosamine (UDP-GlcNAc) and MnCl₂ were added during purification. I can assume that you would expect to see them in the catalytic centers of the GT domains. Are they present there?

9. Line 346. You have successfully solved high-resolution structures and you did not need to run special software for flexible proteins such as cryodrgn or 3DFlex. Why do you claim that the protein has intrinsic flexibility?

Reviewer #3

(Remarks to the Author)

In this manuscript, the authors presented the first structure of the lymphostatin LifA from an enteropathogenic *E. coli* determined using cryo-EM. Data processing of a data set collected at pH 4.0 reveals an L-shaped structure - consistent with previously documented low resolution studies, but the much higher resolution obtained here provides substantial insights into structure and potential functional mechanisms. The authors isolated two distinct structural conformations from their dataset (denoted conformations I and II), which reflect a rearrangement of the delivery domain at the C terminus. The two conformations feature different networks of stabilising hydrogen bonds. When separately resolved, the maps for conformations I and II achieved resolutions of 2.7 Å and 2.8 Å, respectively, permitting accurate modelling and resolving (for the first time) the domain architecture of lymphostatin, showing a total of six domains connected by five linkers.

The authors propose that recombinantly purified LifA is inactive, on the basis that one of the two glycosyltransferase domains lacks the canonically required DXD motif that is necessary for activity, while the other has a DXD motif in its active site, but has a substrate binding site occluded by the linker domain connecting the two GT domains. Likewise, the protease domain, which appears to retain a functionally active catalytic site, but this site is occluded by a loop, the positioning of which may be regulated by two prodomain-like helices.

The authors propose that this inactive LifA is still capable of binding to plasma membrane and subsequently being internalised. They present live cell fluorescent microscopy experiments which show adherence of toxin in foci at the plasma membrane, with internalisation observed in HEK293 cells after 15 hours.

There are no major issues with the experimental methodologies. Overall, the manuscript is well written and easy to comprehend. The manuscript primarily describes the structure of this novel and interesting toxin, with no functional or mechanistic data supporting the hypotheses drawn from the structure, with the exception of the live cell imaging data which simply shows the toxin is internalised. Therefore, the novelty and significance of the manuscript largely rests on the novelty of the structure.

Suggested improvements

The conclusion in the abstract that LifA has "more functional domains than any other known large bacterial toxin" and "can be regarded as the multifunctional Swiss army knife of pathogenic *Escherichia coli*, enabling complex interactions with the host cells in different environments." seems unfounded - most certainly in terms of what is presented in the paper. There is no demonstration that any of the domains are functional, and there are many examples of other bacterial toxins that contain many potentially functional domains (as an example, see the structure of the mcf toxin published by Belyy et al. in *Nature Comms* last year doi:10.1038/s41467-023-44069-2). It is my opinion that LifA is not exceptionally rare or distinct in this regard and the statement should be removed.

The authors conclude that there are no differences between the conformations I and II at pH 4.0 and 6.5, but this analysis does not appear to be very rigorous or quantitative. How was this conclusion reached? Where the two maps quantitatively compared (e.g. using an FSC test?) It would appear that the pH 6.5 structures should be of sufficient resolution to permit model building, and given that structures have already been built into the pH 4.0 maps, refining these into the pH 6.5 maps should be somewhat straightforward. These models could then be quantitatively compared. I would recommend the

authors deposit both structures, regardless of how similar they are.

Line 169-170. "At pH 4.0 and 6.5 both conformations were present (Extended Data Figure 2). This differs from TcdA and TcdB LCTs, which transition from a compact form into an elongated form when the pH is lowered." - Even though there are minimal apparent differences in conformation between the two pHs based on the resolved maps, did the authors consider or investigate whether there is a preference for a particular conformation? In Extended Data Table 1, the map of pH 4.0 conformation I is made up of ~400k while conformation II map is made up of ~800k. In pH 6.5 both conformations showed similar particle numbers. Could this mean that there is a preference for conformation II at lower pH? Is it additionally possible there are other contributing factors in vivo that would push the equilibrium further in favour of either conformation at the two different pHs? Did the authors explore any other pHs?

The functional significance of the two conformations can and should be discussed more thoroughly. It seems unusual that the toxin would exist in two structurally distinct but functionally equivalent conformations. Since the glycosyltransferase and protease domains are blocked in both conformations, the conformational change seems unlikely to contribute to inactivation of the toxin, so what is the functional significance of these? There is limited analysis or discussion of the ART domain - a potential toxic effector which is found within the arm that "moves" between the two conformations. Is the ART domain functional? Is it possible to evaluate mutations within this domain if the functional significance is unknown?

In the live cell imaging experiments, how can the authors be sure that the toxin taken up by the cells represents one, either or both of the conformations characterised structurally? The conclusion that the inactive toxin represents a form that clusters on the membrane prior to internalisation rests on the assumption that the conditions in the culture are the same as those used for cryo-EM imaging. How was this controlled for? The cells appear to have been maintained in PBS which is a higher pH than the imaging conditions, and the protein was conjugated with fluorophore at pH 8. Was this fluorescently labelled protein ever analysed by EM to look for conformational differences?

Map resolution in Extended Data Table 2 (Conf 2, 2.8Å) and overall resolution in Extended Data Table 1 (2.5Å) is different?

Line 354 - Cryosparc live should be Cryosparc live.

The authors provide a very detailed description of the image processing workflow in the methods, which is fantastic and should always be encouraged. I would highly recommend that they consider including a graphical summary of this as an additional supplementary figure.

Reviewer #4

(Remarks to the Author)

Version 1:

Reviewer comments:

Reviewer #1

(Remarks to the Author)

[No comments for author]

Reviewer #2

(Remarks to the Author)

The authors have very thoroughly responded to the reviewer comments, including my own. I think the paper is substantially strengthened by the additional biochemical and structural data. This is a great piece of work, congratulations to the authors. I recommend accepting the article without further delay.

Dear Reviewers,

We thank you for your thorough and constructive reviews. We understand that you have very similar critiques. You all identified the lack of functional assays as a major shortcoming of our manuscript. In particular, you asked for probing the activity of the ART-domain and the activation mechanism of the protease domain. You also agreed in the critique that structural data at higher pH are missing.

In response to your common critiques, we added many more experimental data in our revised manuscript:

1. Structural analysis at pH 8
 - a. Confirms conformation II but not conformation I. Instead, we found a new conformation that we dubbed conformation III.
 - b. Structural comparison of LifA and LifA labelled with Cy3B, showed the same conformations II and III in labelled and unlabelled LifA.
 - c. Mn²⁺ was identified in the catalytic centre of GTI at pH 8 in samples in HEPES buffer but not in phosphate buffer.
2. Functional assays monitored NAD⁺ dependent ADP-ribosylation by the ART domain and showed no activity.
 - a. In vitro tests for self-modification of the ART domain, LifA and LifA + Art domain showed no self-modification under conditions where PARP3 (positive control) shows self ADP-ribosylation
 - b. ART domain or LifA show no NAD⁺ dependent modification of HEK-cell lysates under conditions, where Iota Ia (positive control) modifies the cell lysate.
3. Protease assays of the protease domain and LifA probing the activation mechanisms of the protease
 - a. The protease domain of LifA does not cleave BSA under conditions, where papain is active. Cleavage by the LifA protease domain cannot be activated by IP6 or Arf3, which are common activators in LCTs and Mcf1.
 - b. Autocleavage of LifA is not activated by adding IP6 or Arf3 suggesting a different activation mechanism compared to large LCTs and Mcf1.
 - c. Autocleavage of LifA can be activated by HEK-cell lysate and shows a C-terminal band of approximately 180 kDa, which is absent in a mutant that lacks the catalytic C1480 (LifA^{C1480A}).
4. Fluorescence imaging of HEK cells and T-Lymphocytes showing an uptake of LifA within minutes in both types of cells. Clusters are taken up together with patches of plasma membrane suggesting uptake by endocytosis.

We think that we addressed your comments by the substantial broadening of our investigations and the more detailed discussion. Please find our point-by-point reply below:

REVIEWER COMMENTS

Reviewer #1 (Remarks to the Author):

Griessmann and colleagues describe the structure of lymphostatin from O127:H6 enteropathogenic *Escherichia coli* using single particle cryo-electron microscopy. Lymphostatin (365 kDa) is reported to be an inhibitor of mitogen and antigen activated proliferation of lymphocytes and proinflammatory responses. It has also been reported to mediate bacterial adherence to cells and influences actin nucleation at sites of attachment. Its glycosyltransferase and cysteine protease motifs are required for activity on lymphocytes, but structural information has remained unknown prior to this manuscript.

The structural analysis reveals a multi-domain protein in two distinct conformational states. The domains are connected by long linkers that in some cases occlude access to the active sites. The structures are exciting in that they also reveal the unexpected presence of an ADP-ribosyltransferase (ADPRT) domain fold at the C-terminus and a second glycosyltransferase (GT) domain fold. The structures appear to be of high quality and are expected to be a major advance for the toxin and *E. coli* pathogenesis fields.

While the structure is a significant accomplishment, the manuscript falls short of what one might expect for a high impact publication. The first weakness is in the significant number of questions that are left unanswered. While experimentally answering some of these may arguably fall outside of the scope of this manuscript, others are simple to address and at least merit discussion. The second pertains to the organization; it was hard to appreciate the structural details as key points were split across multiple figures.

We appreciate the concerns of this reviewer. We have reworked the organization of the figures and the text to describe the results in a more structured way.

We have also extended the discussion.

*We acknowledge the shortcoming in functional assays. Other than for many other toxins, cellular targets and mode of action are largely unknown for lymphostatin. So far, there is a general lack of reported *in vitro* assays that demonstrate the functions of the effector domains of lymphostatin.*

In the revision, we added functional assays that probe the activity of the ART domain in comparison to other known ADP-ribosyltransferases such as PARP3 and Iota Ia. We show that the ART domain of LifA is inactive, under conditions where PARP3 and Iota Ia are active.

We also added assays that probe the activation of the protease domain. We show that the known activators INSP6 of LCTs and ARF3 of Mcf1 do not activate the protease in lymphostatin. However, HEK-cell lysate provides the necessary activator that triggers autocatalytic cleavage of lymphostatin.

Major points and unanswered questions.

1. The authors make a point to say that LifA can adopt two different conformations and that this

conformational flexibility does not depend on pH. They only test pH 4.0 and 6.5 though. What happens when the pH is above neutral, eg. pH 8.0?

We followed the suggestion of this reviewer and have collected a large data set at pH 8.0. This showed that conformation 1 had disappeared and a new conformation (conformation 3) had appeared. Conformation 3 showed a large domain reorganization in the C-terminal arm including the helical bundle of the vertex domain. The N-terminal arm with both GT domains and the protease domain did not change in its structure. We have included these new results in the revision.

2. The authors make a point to say that the cysteine protease activity depends on low pH. Why? In the case of the LCTs, zinc is needed to remove an inhibitory zinc. Is a zinc observed in the cysteine protease active site?

The protease domain of lymphostatin is not related to the metalloproteases that are found in RTX toxins or from the Botulinum Neurotoxins. It is also lacking the classical HEXXH motif of this class of proteases that is involved in binding of a Zn ion. In addition, we do not observe a Zn ion in the catalytic site. We have added information that these types of proteases are not related to the protease of lymphostatin in the manuscript. The reason to believe that acidification is required for autocatalytic cleavage is based on an earlier investigation by Bease et al. 2021 in which they show that a 140 kDa cleavage product of LifA is formed in cells depending on the cysteine in the catalytic site of the protease domain and an endosomal acidification. This observation was mentioned in the introduction. Although this observation does not mean that low pH is sufficient to activate the protease, it suggests that endosomal acidification plays a role in lymphostatin processing.

3. The analogy to the pro-domain of a caspase is not well supported. The structural overlap was not clear from the figures. Further, is there any experimental evidence to suggest that this is a mechanism of activation in LifA? The more obvious hypothesis based on the homology to the PaTox/LCT cysteine protease domains is that there is an allosteric activator such as InsP6. Is there any structural evidence to suggest that an allosteric activator could bind another portion of this domain?

It is unknown, what the mechanism of activation of the protease in lymphostatin is. Up to now, protease activity of LifA could not be induced in vitro. Although it is likely that an allosteric activator exists, it is unknown what its nature is.

In LCTs the activating factor is an INSP6, which binds to a positively charged flap. Such a positively charged patch is absent in lymphostatin, making it highly unlikely that the LifA protease is activated by InsP6. Past publications mention that InsP6 could not activate the protease of LifA in vitro (Cassady-Cain et al. 2016 JBC).

We added an extended figure that shows the missing basic flap in comparison to the protease of LCTs. We also added new experimental data that shows no activation of the protease activity of lymphostatin by IP6.

The protease of Mcf1 (makes caterpillars floppy 1) is activated by binding of Arf3 to an activating domain distant from the protease domain. The protease domain of Lifa is indeed very similar. However, the organization of the N-terminal effector domains relative to the protease differ in their topology in Mcf1 and Lifa. We added this comparison with Mcf1 and included an extended figure, which shows the similarities and differences. We also included an assay that tested whether ARF3 can trigger autocatalytic cleavage of lymphostatin in vitro. We report that Arf3 does not activate the protease of lymphostatin. Finally, we showed, that HEK-cell lysate can activate autocatalytic cleavage of lymphostatin. However, we do not know, which component of the lysate is the activator.

4. The authors make a point to say that the active sites of GT-I and the cysteine protease domain are occluded. Are the sites equally occluded in both structural states? If not pH, can they speculate on what will be needed to induce an accessible and active enzyme?

In all three conformations, the accessibility to the catalytic sites of GT-I and the protease domain are unchanged. A release and unbinding of the linkers will be required for priming the structural reorganization. We have noted that the new conformation III could precede activation of autocatalytic cleavage as it brings the delivery domain and the N-terminal helix of the protease domain into contact ready for pulling linker L-II into the catalytic cleft.

This is now added to the description of the delivery domain and discussed.

5. The authors report a C-terminal ADPRT fold. Are the key conserved residues for ADPRT activity present? Does the domain bind NAD? This is easy to test.

We have added functional assays. The ART-domain has a canonical R-T-E motif. However, all residues are shifted and cannot form an active site. The potential NAD⁺ binding site is occupied by the C-terminus of Lifa suggesting another occluded binding cleft.

We have added more details that show the distorted catalytic triad in comparison to Vis toxin, which was identified by FoldSeek.

We noticed that it is not so easy to test NAD⁺ binding. The related ViS toxin has a K_M at some 270 μM . We do not obtain suitable concentration of Lifa and the purified ART domain to test occupancy with ITC or fluorescence anisotropy. However, we have added assays which show no ADP-ribosyltransferase activity of the ART domain or Lifa under conditions where PARP3 or the Iota Ia toxin are active. These assays are added to the manuscript, and we now discuss the ART domain as likely to be catalytically inactive.

6. Lines 98 and 108. Given the strong structural homology between the TcdA and TcdB GT domains, it is potentially misleading to imply a distinction where GT-I is more like TcdA GT and GT-II is more like TcdB GT. Is there a biological reason to make this distinction?

The statement was based on the first hit of a GT domain with known structure by FoldSeek. We agree that this can be misleading and other GT domains with known structure are also identified with a probability of 1 in being a structural homologue. We have rephrased the statement, and we now mention also other GT domains which have been found. To demonstrate the structural similarity, we added a supplementary figure, which shows GT-I and GT-II in comparison to the structural homologues.

Organizational points:

We fully agree with this reviewer that our manuscript had many organizational shortcomings. We have reworked many figures and paid special attention that the numbering of the (extended) figures has the same order as they appear in the text. Consequently, we have also moved contents between figures and generated new figures to place them in a more logic order in the text.

In many places, it would help to have simpler, larger, and higher resolution figures. (For example, Figure 1 is great.) I recommend that the authors present their figures in the order that they present the text describing the results. Please explain the content in the extended data.

We extended figure 1 by the new conformation III, which we discovered at pH 8 when performing additional experiments during the revision. We kept the colour code and show more views of map and model in the extended figure. We took out panel B with the interactions as we will discuss it later in the text. However, we included a presentation of all the domains in panel B as we introduce them in this paragraph.

Line 67 – Extended Data Figure 1 statement is unclear what it is referring to in the panel. Likely panels F & G. If so, move them so they're panels A & B so the figure is in the order in which they are referred to in the manuscript. Explain the other panels.

Extended data figure 1 has been renumbered to extended data figure 2. We reworked the figure, and it now includes only the data from a typical purification. As buffer conditions were identified by a thermal shift assay, we included some selected results before as extended data figure 1.

Lines 102 – 108 – The text description does not match the figure. There are no annotated residues for occupied loops (residues 532-542, figure 2a) or to the referred linker L-1 into the binding cleft.

This is true, the figure only highlights L1, we took out the cross-reference to Figure 2A.

Line 129 – Coming back to figure 2 is confusing. I would have Figure 2 focused on the glycosyltransferase fold and devote Figure 3 to the protease site. This would allow the authors to modify the close-up orientations in Figures 3b-e to allow the reader to understand where the described residues are located. Currently, the close-ups differ from the overview.

We have regrouped the figures, with figure 1 devoted to the three conformations and overview of all domains. Figure 2 is focussed on both GT domains, figure 3 on the protease domain and figure 4 on the vertex domain. We also added a figure 5 for the ART domain. We have extended the figures 3 (protease) and 5 (ART) by the functional assays.

The rationale for transitioning to Figure 4 was not obvious and did not add much in my view.

We think that it is essential to show that LifA as we purify it stably attaches to the cells and is eventually taken up. Neither uptake nor attachment has been shown before in life cell imaging. Following the suggestion of the other reviewers, we have extended the fluorescence imaging to bovine T-lymphocytes and repeated the experiments on HEK cells in cell growth medium. This shows attachment and an uptake into cytosolic clusters, which we think are endosomal compartments for both types of cells with similar kinetics. This demonstrates that the purified protein is taken up in the absence of a type 3 secretion system. The uptake as small, isolated clusters together with plasma membrane is typical for entry via endocytosis, rather than an autonomous translocation.

We have performed further experiments, which show that the Cy3B-labelled LifA adopts similar conformations as the unlabelled enzyme, and that it inhibits T-Lymphocyte proliferation similarly effective as reported earlier.

Other points:

Lines 25-27: It is misleading to say that LifA has more functional domains than any other bacterial toxin. No evidence to suggest GT-II and ADPRT domain are functional and RTX toxins can have five or more enzyme activities.

We added the information on RTX domains and have removed the claim of "more functional domains than any other".

Line 44. References 8/9 do not seem appropriate for this statement.

We replaced the references with two reviews (Orrell et al. 2021 and Kordus et al. 2022).

Extended data Figure 2. It would be helpful to know how panel a relates to the orientations shown in b.

We have removed panel a from the manuscript, because we thought it does not add much. The figure is renumbered to extended data figure 3.

Line 98 – Why skip to “Extended Figure 4a (sic)” when Extended Data Figure 3 has not been described?

We fixed this.

Extended Data Figure 3 is not talked about in the text. While nice to have, it would be helpful if panels were larger with less white space. Same with Extended Figure 4. Extended data figure 5 is great.

Extended Data Figure 3 has now moved into panel b of figure 1 with the white space removed. It is cross referenced, when all the domains and linkers are mentioned.

Extended Data Figure 4 has been removed. The contents went to Figure 2 for the close-up of the active site of the GT-domains and Extended Data Figure 5 for the comparison of different GT-domains. The protease domain representations are now in Figure 3 and Extended Data Figure 9.

Extended Data Figure 5 has also been reworked to accommodate conformation III. It is now Extended Data Figure 12.

Figure 4 and Extended data Figure 6. Recommend changing colors to accommodate red/green colorblind individuals. Gray scale for individual channels is easiest to see.

We changed the colouring and replaced the data with the new up-take experiments in medium.

Minor points:

Line 101 – Add comma after LifA

done

Reviewer #2 (Remarks to the Author):

Lymphostatin is a virulence factor of pathogenic *E. coli*, whose molecular mechanism of action remains largely unclear. In this manuscript, the group of Bettina Böttcher determined two high-resolution structures of this large bacterial toxin and described its architecture at the molecular level. The results obtained are original and significant for the field. The manuscript contains structural data of outstanding quality, from which the authors draw far-reaching conclusions about the functionality of the toxin without any additional biochemical studies. The latter issue needs to be addressed before publication.

We appreciate that it would be nice to understand what the molecular mechanisms of lymphostatin are. However, many efforts along these lines by several researchers in the past 20 years have failed to reveal the activity of LifA *in vitro*. Many of the experiments were guided by the wealth of existing information on LCTs and the assumption that lymphostatin could act similarly. However, none of those assumptions could be experimentally confirmed. Therefore, molecular targets and functions are largely unknown. This makes the design of activity tests to a fishing expedition with unclear outcome.

In response to the reviewer, we have added biochemical studies to address selected aspects of the functionality. In particular, we tested for ADP-ribosyltransferase activity and activation of autocatalytic cleavage by IP6 or Arf3. This is further detailed below in the response to the reviewer.

1. The authors state that lymphostatin has the highest number of domains of any known toxin. I am surprised by this claim since Multifunctional-autoprocessing repeats-in-toxin (MARTX) toxins typically have 5+ effectors and several other domains necessary for efficient delivery and function. For review: <https://doi.org/10.1128/microbiolspec.ve-0002-2014>.

We have removed the claim.

2. Line 82: The authors describe the identification of an ADP-ribosyltransferase domain at the C-terminus of the toxin. This is a very interesting discovery. But what is the evidence that this domain has this particular function? The structure itself is not sufficient for such a statement. Please provide the results of functional experiments.

The assignment is based on the structure, which we would consider being sufficient for grouping a protein into a protein family. Some designated PARPs have been shown to be inactive.

We agree that belonging to a protein family does not proof a certain function. We are now more explicit on the assignment based on structural similarity. The canonical R-S/T-E motif does not form the expected triad in the ART domain. We added functional assays that show inactivity of the ART domain and of LifA under conditions where Iota Ia is active. We discussed the lack of structural preservation of the catalytic centre in more detail.

3. Line 108. Is the structural data the only evidence that GT-2 is a glycosyltransferase? I am skeptical of this claim, especially because this domain lacks a common DXD motif. Please perform functional studies to show that the GT-2 domain performs this function.

As the reviewer pointed out, we have based the assignment purely on the structural similarity and the probability of 1 given by FoldSeek that this fold is a structural homologue to the glycosyltransferase of several toxins. To illustrate the similarity, we added the Extended Data Figure 5a, which shows the side-by-side presentation with the structural homologues. DXD motifs are frequently observed in glycosyltransferases, however, they are not strictly conserved. We added this information that the DXD motifs are not strictly conserved. We also added the information that the assignment of the domain to the protein family is based on the fold and that we do not know whether GT-II is functional or not.

We appreciate the wish for further functional tests. However, it is unknown how the glycosyl transferase activit(ies) in lymphostatin can be activated and what their cellular targets are. Therefore, it is less than trivial to demonstrate its activity, and we failed to establish a suitable assay.

4. Line 127. The unavailability of the protease domain is also asserted based on structure alone. TcdA and TcdB toxins are activated by inositol phosphate cofactors <https://www.nature.com/articles/nature05622> and Mcf-like toxins are activated by Arf proteins <https://www.pnas.org/doi/abs/10.1073/pnas.1905095116> I suggest that the authors test whether the addition of these molecules activates protease activity.

The protease is lacking a basic flap for binding inositol phosphate co-factors. Bease et al. and we did not observe autocatalytic cleavage when adding IP6. We repeated this experiment testing IP6, we also tried to activate the protease by adding purified ARF3,

which activates the protease in Mcf1. We did not observe an activation. We also added HEK-cell lysate and showed that it is sufficient to activate auto catalytic cleavage. We also tested whether the isolated protease domain in the absence of sterical hindrance by the surrounding domains could cleave BSA. However, we did not observe activity of the protease domain under conditions where Papain is active. The outcome of these assays is now added to the results.

5. If both conformations I and II are present at pH 4 and 6.5, what was the rationale for determining their structures at both pHs? Are they physiological states or could one of them be an artifact in protein production? I suggest that the authors measure the LD50 of the toxin on cell cultures and compare it with previous reports.

The rationale of starting with pH 6.5 was based on the increased thermal stability seen in a thermal shift assay. As we observed two conformations, we reasoned that we might not be acidic enough for converting all LifA into a potential low pH form. Therefore, we acquired data at pH 4 to enrich the low pH-form. Comparisons of the ratios of conformation I and II at pH 4 and 6.5 did not show a significant change in ratios that we could confidently attribute to the pH shift. This rationale is now elaborated in more detail in the results section.

Following the suggestions of all three reviewers, we also added structure determination at pH 8.0 of LifA shifted to pH 8.0. This shows that conformation I disappeared suggesting that it is indeed the low pH form. In addition, a new conformation appeared, which we call conformation III. This suggests some pH dependence of conformation I as the low pH form and conformation III as the high pH form.

Lymphostatin inhibits the mitogen activated lymphocyte proliferation, but it has no cytotoxic or necrotic effect on the cells. We added this information to the introduction. In the absence of a cytotoxic effect the determination of the LD50 becomes obsolete.

Following the suggestion by this reviewer we tested the effectiveness of the dye-labelled lymphostatin in inhibition of lymphocyte proliferation and found a similar potency as reported previously and also similar for labelled and unlabelled LifA.

6. Line 185. I am confused by the last statement of this paragraph. Bacterial toxins are known to act quickly and efficiently, for example, C. difficile toxins A and B are internalized within 20-30 minutes. If the toxin is functional, it should not take 15 hours for it to penetrate target cells.

We have revisited this issue and repeated the experiments with HEK cells and with T-lymphocytes. For the live cell imaging of HEK cells we omitted the starvation in phosphate buffer and imaged the cells in medium. Indeed, for both cell types we observe an uptake into cytosolic foci within minutes. In the revision, we replaced the original experiment

showing adherence of LifA to HEK cells in phosphate buffer with the new experiment showing uptake of LifA by HEK-cells in medium. The live cell imaging also shows that LifA-clusters first attach to the cell membrane before they are taken up together with cell membrane hinting at an endocytic pathway.

We would reconcile the previous experiment as the cells were significantly affected by the starvation in PBS buffer in the flow chamber with a subsequent down regulation of endocytosis. Probably by replenishing the growth medium for the over-night incubation, endocytosis normalized and the tightly adhering LifA was taken up.

7. Line 243. If the data suggest T3SS-mediated toxin entry, why does the toxin need a delivery domain? Please discuss.

Delivery domain does not necessarily mean autonomous translocation but also receptor binding. LifA acts on lymphocytes in the absence of bacteria and secretion systems. The light microscopy suggests an uptake by endocytosis, the attachment hints at a receptor binding.

T3SS can inject effectors into the target cell but also translocate effectors into the medium. LifA might act differently depending on the route by which it is administered. This is mentioned in the discussion.

The activity against mitogen activated lymphocyte proliferation is independent of T3SS.

8. The protein expression protocol has several interesting elements that are not adequately discussed.

8.1. Why did you purify the protein at pH 6.5 in the first place?

See answer to point 5 of the same reviewer. This is based on the stability of the thermal shift assay.

8.2. (1-Pyridine)-1-Propanesulfonate was added to stabilize the protein. How did it help? Could it have affected the structure of the protein, leading to the appearance of two conformations?

(1-Pyridine)-1-Propanesulfonate is a common additive in protein purification for stabilizing the fold of proteins. The addition of PPS was already described for the purification of LifA by Cassady-Cain 2016, which we adapted. We added the reference to the materials and methods. We kept many of the initial buffer conditions, including the PPS. After the IMAC, PPS is no longer present. As we now have a conformation III at pH 8 and lost conformation I, we think it is unlikely that PPS leads to the two conformations.

8.3 Line 288-289. Uridine diphosphate N-acetylglucosamine (UDP-GlcNAc) and $MnCl_2$ were added during purification. I can assume that you would expect to see them in the catalytic centers of the GT domains. Are they present there?

We do not observe density that we can confidently attribute to UDPGlcNAc or Mn^{2+} in the maps of lymphostatin purified in phosphate buffer at pH 4 or pH 8. However, not seeing a density is not always proof of being absent and might simply represent larger mobility or incomplete occupancy. Generally, the resolvability in this region close to the DXD motif in the GT-I domain is not as good as we would hope for and there is some disconnected density, which we can attribute to parts of linker L1. However, despite these limitations, we were puzzled that we did not observe a strong density for a bound Mn^{2+} . We reasoned that this was due to the low solubility of Mn^{2+} in phosphate buffer. During the revision, we acquired part of the new data set of lymphostatin at pH 8 in HEPES buffer instead of phosphate buffer, both with added Mn^{2+} . In the data set of LifA in HEPES, we could resolve a density for Mn^{2+} but not in the data of LifA in phosphate buffer. We added this information to our revision.

The GT-II domain is better resolved close to the EEN motif compared to the GTI domain close to the DXD-motif. Here, we observe some low-density speckles. However, we consider them to be too low density to account for a Manganese ion. Comparison to the DXD motif in TcdA with bound Mn^{2+} and UDP (4DMW) suggests that the speckles are also at the wrong place for an expected Manganese ion. All in all, this assignment is too speculative to discuss it in more depth in the manuscript.

The GT-II domain in EM-density at pH 4.0. The EEN domain is shown in green and a coordinated density (blue) is close by.

9. Line 346. You have successfully solved high-resolution structures, and you did not need to run special software for flexible proteins such as cryodrgn or 3DFlex. Why do you claim that the protein has intrinsic flexibility?

We tried 3D Flex. However, we could not see significant improvements. We found that 3D Flex was not able to solve the problems of large dynamics. For us, the better approach

was to do local classifications without alignment of the individual parts and then continue with the best class. For the very problematic C-terminal region downstream of Gly2630 the 2D-class averages showed little overlap, which also explains why the tip of conformation II is so badly resolved in a consensus map.

So, our special approach was not cryo-Drng or FlexEM but using a subset of particles, which represents a domain of interest in the most consistent way. This is only possible, because we started with very large data sets.

We also tried Multibody refinement in Relion. This worked better than FlexEM and gave a good appreciation of the relative domain movements. However, the best results were obtained by focussed classification and refinement.

I guess there are different routes that work for overcoming flexibility problems. By now, we have collected some 100,000 movies of LifA and completed some 1000 jobs with RELION and cryoSPARC, Flex EM, multibody various classifications, variability analyses and heterogenous refinements. We ended up with the old-fashioned local classification and local alignment to solve our problem.

Reviewer #3 (Remarks to the Author):

In this manuscript, the authors presented the first structure of the lymphostatin LifA from an enteropathogenic *E. coli* determined using cryo-EM. Data processing of a data set collected at pH 4.0 reveals an L-shaped structure - consistent with previously documented low resolution studies, but th much higher resolution obtained here provides substantial insights into structure and potential functional mechanisms. The authors isolated two distinct structural conformations from their dataset (denoted conformations I and II), which reflect a rearrangement of the delivery domain at the C terminus. The two conformations feature different networks of stabilising hydrogen bonds. When separately resolved, the maps for conformations I and II achieved resolutions of 2.7 Å and 2.8 Å, respectively, permitting accurate modelling and resolving (for the first time) the domain architecture of lymphostatin, showing a total of six domains connected by five linkers.

The authors propose that recombinantly purified LifA is inactive, on the basis that one of the two glycosyltransferase domains lacks the canonically required DXD motif that is necessary for activity, while the other has a DXD motif in its active site, but has a substrate binding site occluded by the linker domain connecting the two GT domains. Likewise, the protease domain, which appears to retain a functionally active catalytic site, but this site is occluded by a loop, the positioning of which may be regulated by two prodomain-like helices.

The authors propose that this inactive LifA is still capable of binding to plasma membrane and subsequently being internalised. They present live cell fluorescent microscopy experiments which show adherence of toxin in foci at the plasma membrane, with internalisation observed in HEK293 cells after 15 hours.

There are no major issues with the experimental methodologies. Overall, the manuscript is well

written and easy to comprehend. The manuscript primarily describes the structure of this novel and interesting toxin, with no functional or mechanistic data supporting the hypotheses drawn from the structure, with the exception of the live cell imaging data which simply shows the toxin is internalised. Therefore, the novelty and significance of the manuscript largely rests on the novelty of the structure.

Suggested improvements

The conclusion in the abstract that LifA has "more functional domains than any other known large bacterial toxin" and "can be regarded as the multifunctional Swiss army knife of pathogenic Escherichia coli, enabling complex interactions with the host cells in different environments." seems unfounded - most certainly in terms of what is presented in the paper. There is no demonstration that any of the domains are functional, and there are many examples of other bacterial toxins that contain many potentially functional domains (as an example, see the structure of the mcf toxin published by Belyy et al. in Nature Comms last year doi:10.1038/s41467-023-44069-2). It is my opinion that LifA is not exceptionally rare or distinct in this regard and the statement should be removed.

We have removed the statement and added comparisons regarding Mcf1, with which the LifA protease domain shares the protein family. We also included a functional assay to test whether the protease domain of LifA can be activated similarly as described for Mcf1.

We also added mentioning of MRTX toxins, which can have similarly many domains but otherwise share little similarity in the domain architecture.

The authors conclude that there are no differences between the conformations I and II at pH 4.0 and 6.5, but this analysis does not appear to be very rigorous or quantitative. How was this conclusion reached? Where the two maps quantitatively compared (e.g. using an FSC test?) It would appear that the pH 6.5 structures should be of sufficient resolution to permit model building, and given that structures have already been built into the pH 4.0 maps, refining these into the pH 6.5 maps should be somewhat straightforward. These models could then be quantitatively be compared. I would recommend the authors deposit both structures, regardless of how similar they are.

The similarity refers to the overall domain arrangement with the ART domain either in the centre or at the distal end of the C-terminal arm. Otherwise, both conformations show mobility in the C-terminal arm. The mobility is documented by the low occupancy/density, which reflects mobility rather than degradation. Therefore, even within one pH class somewhat different positions of the arms in respect to each other emerge without effecting the overall domain arrangement.

The quality of the maps at pH 6.5 is much lower than those at pH 4.0. The data for the maps at pH 6.5 were collected with an older set-up with a Falcon III camera. For speed reasons, the images were collected in linear mode and with an image shift that was not compensated for beam tilt during acquisition (only later during processing). The primary image data is worse than for the later data sets, which were collected with a Falcon IVi and Selectris filter in counting mode.

All maps suffer from an apparent preferential orientation. This apparent preferential orientation leads to a wrong assignment of the Euler angles, which ends up with a low precision and a distorted map. Typically, we overcome this problem, by identifying subsets in classification and heterogeneous refinement that form less distorted maps. For the linear data, which has only half of the DQE of counted data, this worked less well. Therefore, "quick refinement" of the model into the map generates distorted models that we do not wish to deposit. However, in response to the request, we have deposited the maps of conformation I and II at pH 6.5.

In the revision, we have added another data set at pH 8 with the new set-up. Here we identified a new conformation. We have deposited these maps and models.

Line 169-170. "At pH 4.0 and 6.5 both conformations were present (Extended Data Figure 2). This differs from TcdA and TcdB LCTs, which transition from a compact form into an elongated form when the pH is lowered." - Even though there are minimal apparent differences in conformation between the two pHs based on the resolved maps, did the authors consider or investigate whether there is a preference for a particular conformation? In Extended Data Table 1, the map of pH 4.0 conformation I is made up of ~400k while conformation II map is made up of ~800k. In pH 6.5 both conformations showed similar particle numbers. Could this mean that there is a preference for conformation II at lower pH? Is it additionally possible there are other contributing factors in vivo that would push the equilibrium further in favour of either conformation at the two different pHs? Did the authors explore any other pHs?

We were also very much interested in the ratios of conformation I and II (and later III) at the different pH values and under different conditions. However, it turned out that this simple question is much more difficult to address than anticipated. The ratio depended heavily on when during the processing the numbers were assigned. Conformation II suffered more from (pseudo)-preferential orientation than conformation I and III. Consequently, more particles were kicked out during the processing. We also tried to assign the ratios very early in processing after in a heterogeneous refinement after the 2D-classification. However, here many miss-picks and none-aligning particles affected the numbers of particles per class.

So, we did not see a clear tendency in the change of ratios. However, conformation II is the most abundant conformation at all pH values. We added a statement along these lines in the results.

We followed the advice of all three reviewers for testing other pH values during the revision and tested pH 8. We did not find particles in conformation I. Instead, we discovered a new conformation, which we dubbed conformation III. From this observation, we concluded that conformation I is the low pH form and conformation III the high pH form. We do not know where to place conformation II as it is present at all pH values and could be either the intermediate pH-form or a trapped conformation that

does not respond to changes in the pH. This new data is added to the manuscript and discussed along these lines.

The functional significance of the two conformations can and should be discussed more thoroughly. It seems unusual that the toxin would exist in two structurally distinct but functionally equivalent conformations. Since the glycosyltransferase and protease domains are blocked in both conformations, the conformational change seems unlikely to contribute to inactivation of the toxin, so what is the functional significance of these? There is limited analysis or discussion of the ART domain - a potential toxic effector which is found within the arm that "moves" between the two conformations. Is the ART domain functional? Is it possible to evaluate mutations within this domain if the functional significance is unknown?

In the revision we tested the functionality of the ART domain and found that it is inactive under conditions where Iota Ia or PAPR3 are active. We also added more discussion and comparison around the ART domain. It appears that the catalytic motif (R-T-E) is present, but the three residues are shifted and do not come together to form a catalytic centre. Furthermore, the potential NAD⁺ binding site is occupied by the C-terminus of LifA in all three conformations. The ART-domain contacts a different site at LifA in all three conformations. We think that this could be related to some sort of self-processing, probably by drawing host factors to a specific site of LifA. We added this data an interpretation to the discussion and the results.

The functional readouts of lymphostatin's function and the general knowledge on how LifA acts and what it targets are limited. Therefore, it is difficult to test this somewhat fuzzy readout by mutations.

In the live cell imaging experiments, how can the authors be sure that the toxin taken up by the cells represents one, either or both of the conformations characterised structurally? The conclusion that the inactive toxin represents a form that clusters on the membrane prior to internalisation rests on the assumption that the conditions in the culture are the same as those used for cryo-EM imaging. How was this controlled for? The cells appear to have been maintained in PBS which is a higher pH than the imaging conditions, and the protein was conjugated with fluorophore at pH 8. Was this fluorescently labelled protein ever analysed by EM to look for conformational differences?

We appreciate the concern that we do not know how labelling changes LifA and approached this question in the revision. Here, we repeated and extended the cell imaging of HEK cells and added imaging of T-lymphocyte treated with labelled LifA.

From this prep of LifA we analysed labelled and unlabelled LifA at pH 8 by cryo-EM and found conformation II and III in LifA and LifA-Cy3B. In addition, LifA and LifA-Cy3B were similarly efficient in inhibiting mitogen activated T-cell proliferation.

We conclude that labelling does not affect the conformation of LifA and its potency in inhibiting Lymphocyte proliferation. We added these experiments to the results. We also removed suggestions that the conformations in the clusters on and in cells are in any way the same as, in the labelled LifA that we add to the medium. Who knows, what LifA does when it binds to a cellular receptor.

Map resolution in Extended Data Table 2 (Conf 2, 2.8Å) and overall resolution in Extended Data Table 1 (2.5Å) is different?

These numbers were correct. Table 1 is at the end of the consensus processing with Cryosparc. Table 2 is the summary of the composite map and estimated from the combined half-maps.

However, we appreciate the confusion caused by the complex processing. Therefore, we have reworked table 1 and 2. In table 1 we summarize the data acquisition and in table 2 information describing the deposited maps and models. We have extended both tables to include the data at pH 6.5 and pH 8.0 in HEPES and phosphate buffer to reflect the deposited data.

Line 354 - Cryosparc life should be Cryosparc live.

fixed

The authors provide a very detailed description of the image processing workflow in the methods, which is fantastic and should always be encouraged. I would highly recommend that they consider including a graphical summary of this as an additional supplementary figure.

We followed the advice of this reviewer and added graphical summaries of the workflows (Extended Data Figure 18-26). As the figure legends contain most of the image processing strategies, we have shortened the description in the materials and methods and refer now to the workflows.